# Lipid homeostasis is essential for a maximal ER stress response

Gilberto Garcia[1,2], Hanlin Zhang[1], Sophia Moreno[1], C Kimberly Tsui[1], Brant Michael Webster[1], Ryo Higuchi-Sanabria[2], Andrew Dillin[1]*

[1]Department of Molecular & Cellular Biology, Howard Hughes Medical Institute, University of California, Berkeley, Berkeley, United States; [2]Leonard Davis School of Gerontology, University of Southern California, Los Angeles, United States

**Abstract** Changes in lipid metabolism are associated with aging and age-related diseases, including proteopathies. The endoplasmic reticulum (ER) is uniquely a major hub for protein and lipid synthesis, making its function essential for both protein and lipid homeostasis. However, it is less clear how lipid metabolism and protein quality may impact each other. Here, we identified *let-767*, a putative hydroxysteroid dehydrogenase in *Caenorhabditis elegans*, as an essential gene for both lipid and ER protein homeostasis. Knockdown of *let-767* reduces lipid stores, alters ER morphology in a lipid-dependent manner, and blocks induction of the Unfolded Protein Response of the ER (UPR$^{ER}$). Interestingly, a global reduction in lipogenic pathways restores UPR$^{ER}$ induction in animals with reduced *let-767*. Specifically, we find that supplementation of 3-oxoacyl, the predicted metabolite directly upstream of *let-767*, is sufficient to block induction of the UPR$^{ER}$. This study highlights a novel interaction through which changes in lipid metabolism can alter a cell's response to protein-induced stress.

## Editor's evaluation

In this work, the often-surmised but still-poorly understood connection between lipid metabolism and ER stress was explored. The work using genetic techniques and a variety of parallel approaches, implicates key lipid synthetic pathways in the rise and strength of an ER stress signal. The studies as submitted were strong, and create a compelling case for these new connections between lipid metabolism and cellular stress response.

## Introduction

The cell must monitor protein and lipid quality to maintain cellular homeostasis. Disruptions in protein folding have been implicated in numerous neurodegenerative diseases, while lipid imbalances result in an increased risk for cardiovascular disease, diabetes, and various cancers (*Hartl, 2017*; *Sletten et al., 2018*; *Chaurasia and Summers, 2015*; *Avgerinos et al., 2019*). These two distinct metabolic pathways have been extensively studied independently, yet increasing evidence suggests that examining their relationship to one another may provide novel insights into human health. Individuals with Alzheimer's disease (AD) and Parkinson's disease (PD), diseases known to be associated with increased development of abnormal protein aggregates, also show dysregulation of lipid metabolism (*Yerbury et al., 2016*; *Alecu and Bennett, 2019*; *Kao et al., 2020*). As such, changes in lipid metabolism are now suspected to contribute to the development of these proteopathies through mechanisms that are not fully understood (*Chang et al., 2017*; *Chew et al., 2020*; *Xicoy et al., 2019*). The endoplasmic reticulum (ER) is a major metabolic hub, responsible for the synthesis of secreted and integral proteins, as well as a major portion of a cell's lipids (*Schwarz and Blower, 2016*). The close

*For correspondence:
dillin@berkeley.edu

Competing interest: The authors declare that no competing interests exist.

relationship between protein and lipid quality control within the ER is highlighted by the Unfolded Protein Response of the ER (UPR$^{ER}$), a system capable of sensing and responding to both protein misfolding and membrane lipid disequilibrium (*Metcalf et al., 2020*; *Xu and Taubert, 2021*).

In higher eukaryotes, the UPR$^{ER}$ contains three unfolded protein sensors, each with their own independent signaling pathways. While all three sensors are ER-localized transmembrane proteins with a luminal unfolded protein sensing domain, the most conserved of these pathways is the inositol-requiring enzyme-1 (*ire-1*) branch. The IRE-1 luminal domain is bound by the resident ER HSP70 chaperone (*C. elegans* HSP-4) under basal conditions. Upon protein folding stress, the HSP70 chaperone is titrated away to allow for the interaction of the luminal domain with misfolded proteins, leading to oligomerization and activation of IRE-1's cytosolic RNase domain. Activated IRE-1 then initiates noncanonical splicing of *xbp-1* mRNA from its *xbp-1u* form to its effective *xbp-1s* variant at the ER membrane (*Gómez-Puerta et al., 2022*). The XBP-1s transcription factor is then able to upregulate the cell's UPR$^{ER}$ target genes to increase protein folding, protein turnover, and lipid metabolism to help ameliorate the stress. (*Frakes and Dillin, 2017*; *Adams et al., 2019*).

All three of the UPR$^{ER}$ sensors are also able to sense membrane lipid disequilibrium through domains adjacent to their transmembrane helices. These domains activate the UPR$^{ER}$ independent of proteotoxic stress and the luminal sensing domains (*Volmer and Ron, 2015*; *Halbleib et al., 2017*; *Tam et al., 2018*). This is unsurprising considering the ER's critical role in the synthesis of major lipids including membrane lipids, cholesterol, and neutral lipids (*Lodhi and Semenkovich, 2014*; *Fagone and Jackowski, 2009*). Accordingly, the ER actively regulates the cell's lipid status through changes to lipid synthesis enzymes, transcriptions factors, and interactions with lipid droplets (LDs), organelles tasked with the storage and gatekeeping of surplus lipid stores (*Shimano and Sato, 2017*; *Jacquemyn et al., 2017*; *Olzmann and Carvalho, 2019*).

Utilizing the same stress response sensors for protein and membrane lipid stress suggests an interdependent link between lipid and protein homeostasis within the ER. Ectopic activation of the UPR$^{ER}$ results in changes to lipid metabolism, while changes in sphingolipid, lipid saturation, ceramides, and loss of various lipid enzymes activate the UPR$^{ER}$ (*Volmer and Ron, 2015*; *Tam et al., 2018*; *Imanikia et al., 2019*; *Daniele et al., 2020*; *Contreras et al., 2014*; *Promlek et al., 2011*). Interestingly, both lipids and LDs have been shown to contribute to proteostasis. A sterol pathway localized to LDs is required for clearing inclusion bodies and LDs themselves are necessary for transport of damaged proteins (*Moldavski et al., 2015*; *Vevea et al., 2015*). Whether other lipid pathways in the ER or on LDs are required for an effective UPR$^{ER}$ response to protein stress has not been fully investigated. Here, we performed a genetic screen of LD-associated genes to identify genes whose knockdown affected UPR$^{ER}$ activation in the nematode *C. elegans*. We identified the hydroxysteroid dehydrogenase, *let-767*, as required for both ER lipid and protein homeostasis. Loss of *let-767* resulted in severe defects in UPR$^{ER}$ activation, lipid homeostasis, and ER morphology. Defects in ER morphology could be rescued through lipid supplementation; however, the deficiencies in UPR$^{ER}$ activation were not. Instead, knockdown of the upstream regulator of the *let-767* pathway resulted in a significant recovery of the UPR$^{ER}$ activation. Thus, we propose that loss of *let-767* results in accumulation of fatty acid metabolites which leads to detrimental remodeling of the ER membrane and disruption of ER functions in lipid synthesis and UPR$^{ER}$ induction. Our work highlights a unique cellular interaction in which lipid metabolism negatively impacts ER protein homeostasis. Further understanding of how these two systems influence one another may bring new insights into the mechanisms of protein and lipid disorders.

## Results

### LET-767 is a regulator of the UPR$^{ER}$

To identify novel lipid genes that influence the UPR$^{ER}$, we carried out an RNAi screen under conditions of proteotoxic stress. We utilized the *hsp-4* (*C. elegans* Hsp70/BiP) transcriptional GFP reporter (*hsp-4p::GFP*) to assess the UPR$^{ER}$ induction, and *sec-11* (an ER serine-peptidase) RNAi to generate ER stress and induce the GFP reporter (*Calfon et al., 2002*). In combination with the *sec-11* RNAi, we individually knocked down each candidate gene to perform a double RNAi screen (*Figure 1A*). We focused on proteins associated with LDs instead of general lipid synthesis genes due to the LD's central role in lipid regulation and their contribution to ER proteostasis (*Moldavski et al., 2015*;

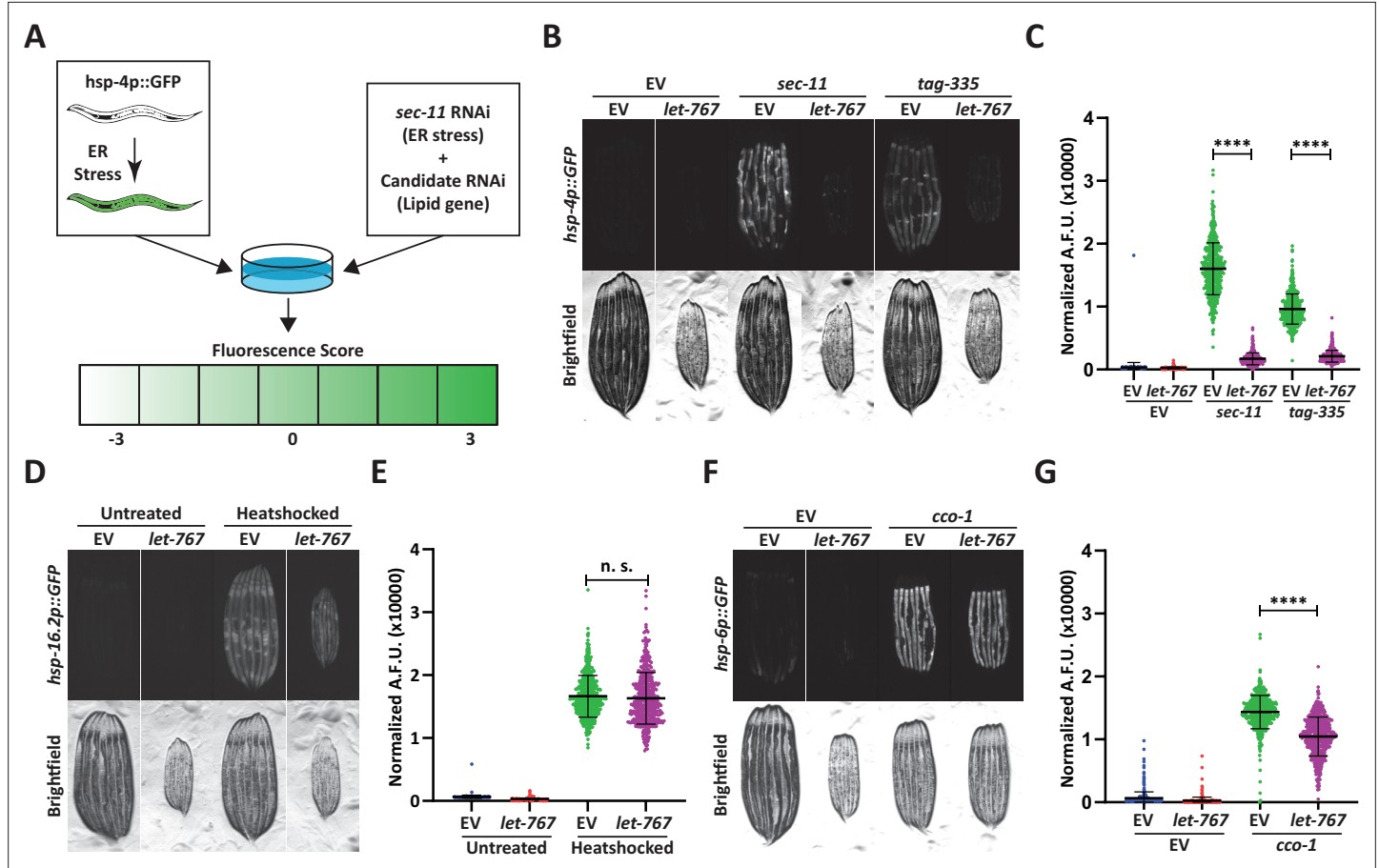

**Figure 1.** Knockdown of *let-767* specifically suppresses the UPR[ER]. (**A**) Schematic for screening method used to identify UPR[ER] modulators from candidate genes. Animals expressing *hsp-4p::GFP* were grown from L1 on candidate RNAi mixed in a 1:1 Ratio with ER stress inducing *sec-11* RNAi. Animals were then screened at day 1 of adulthood and scored for changes in fluorescence compared to the *sec-11*/Empty Vector (EV) control. (**B**) Fluorescent micrographs of day 1 adult transgenic animals expressing *hsp-4p::GFP* grown from L1 on EV, *sec-11*, or *tag-335* RNAi combined in a 1:1 ratio with either EV or *let-767* RNAi to assay effects on UPR[ER] induction. (**C**) Quantification of (**B**) normalized to size using a BioSorter. Lines represent mean and standard deviation. n=500. Mann-Whitney test p-value ****<0.0001. Representative data shown is one of three biological replicates. (**D**) Fluorescent micrographs of day 1 adult transgenic animals expressing *hsp-16.2p::GFP* grown from L1 on EV or *let-767* RNAi with or without 2 hr 34 °C heat-shock treatment to assay heat shock response. Animals imaged 2 hr after recovery at 20 °C. (**E**) Quantification of (**D**) normalized to size using a BioSorter. Lines represent mean and standard deviation. n=400. Mann-Whitney test n.s.=not significant. Representative data shown is 1 of 3 biological replicates. (**F**) Fluorescent micrographs of day 1 adult transgenic animals expressing *hsp-6p::GFP*, grown from L1 on EV or *cco-1* RNAi combined in a 1:1 ratio with either EV or *let-767* RNAi to assay effects on UPR[mt] induction. (**G**) Quantification of (**F**) normalized to size using a BioSorter. Lines represent mean and standard deviation. n=431. Mann-Whitney test p-value ****<0.0001. Representative data shown is one of three biological replicates.

The online version of this article includes the following figure supplement(s) for figure 1:

**Figure supplement 1.** Knockdown of *let-767* suppresses the UPR[ER] more severely than other lipid related genes.

*Vevea et al., 2015*). By cross-referencing two independent *C. elegans* LD proteomic datasets, we identified 163 high-confidence LD proteins (*Zhang et al., 2012*; *Na et al., 2015*; *Vrablik et al., 2015*). We found that RNA interference of 49 genes led to reduced induction of the UPR[ER] reporter (*Supplementary file 1*). While a large portion of these genes are annotated as functioning in general translation (e.g. ribosomal subunits), potentially affecting global gene expression, a subset of 11 belonged to other functional groups and were therefore more likely to specifically affect the UPR[ER] (*Figure 1—figure supplement 1A*). From this subset of genes only *let-767*, an acyl-CoA reductase, was noted as having lipid enzymatic activity, while the other genes function as ATP synthases, GTPases, heat shock proteins, ATPases, methyltransferases, and tubulin.

We focused on *let-767* of the screen subset due to its direct role in lipid metabolism, which has been previously characterized (*Entchev et al., 2008*; *Desnoyers et al., 2007*; *Kuervers et al., 2003*).

*let-767* has been implicated in long-chain fatty acid (LCFA) and monomethyl branched-chain fatty acid (mmBCFA) synthesis as an acyl-CoA reductase, and in steroid metabolism as a 17-beta hydroxysteroid dehydrogenase that directly acts on steroid molecules. Within the fatty acid elongation pathway, after addition of a malonyl-CoA, acyl-CoA reductase is responsible for reduction of the 3-ketoacyl-CoA into a 3-hydroxyacyl-CoA which is then further processed into a fatty acid with 2 more carbon molecules. Knockdown of *let-767* suppressed the UPR^ER induction caused by RNAi of two different ER genes, *sec-11*, which would impact ER signal peptide cleavage, and *tag-335*, a GDP-mannose pyrophosphorylase, which would reduce protein glycosylation (*Figure 1B–C*; *Bar-Ziv et al., 2020*). Furthermore, qPCR analysis confirmed that *let-767* RNAi also resulted in reduced splicing of *xbp-1u* to *xpb-1s* while under stress, suggesting that loss of *let-767* suppresses the UPR^ER induction by preventing IRE-1's splicing of *xbp-1* (*Figure 1—figure supplement 1B*). Next, we utilized the transcriptional reporters of the mitochondrial unfolded protein response (UPR^mt, *hsp-6p::GFP*) and the heat shock response (HSR, *hsp-16.2::GFP*) to determine the effect of *let-767* knockdown on other stress responses. We observed that *let-767* RNAi does not suppress the heat shock response and only has minor effects on the mitochondrial stress response in comparison to the highly suppressed UPR^ER (*Figure 1D–G*). Together, these results show that the function of *let-767* is specifically important for ER function and homeostasis.

Both fatty acid elongation and steroid processing is carried out by steroid hydrogenases within mammals (*Sakurai et al., 2006*). To determine whether reduced UPR^ER induction was a general phenotype of steroid dehydrogenase knockdown, we performed our double RNAi protocol on the four most closely related (steroid dehydrogenase family) genes found in *C. elegans*, *stdh-1,2,3* and *4*. Knockdown of any of the *stdh* genes had only mild effects on the UPR^ER induction, showing that the diminished UPR^ER induction phenotype was more specific to *let-767* and not a general phenomenon of decreased steroid dehydrogenase function (*Figure 1—figure supplement 1C–D*). We then tested whether RNAi of other genes implicated in the LCFA and mmBCFA pathways also affected the UPR^ER induction (*Zhang et al., 2011*; *Kniazeva et al., 2004*; *Kniazeva et al., 2012*). Of these genes, *acs-1* had the most similar phenotype to *let-767*, while *elo-5* and *hpo-8* had a significant, but less severe effects on the UPR^ER induction without stalling development, like *pod-2* (*Figure 1—figure supplement 1E–F*). *acs-1* functions in the first step of fatty acid processing by attaching a CoA to the fatty acid, particularly for the mmBCFA isoC17 (*Kniazeva et al., 2012*; *Zhang et al., 2021*). While *elo-5* and *hpo-8* function as a fatty acid elongase and a 3-hydroxyacyl-CoA dehydratase, the steps before and after *let-767* in the fatty elongation pathway, respectively. These results indicate that perturbation of the LCFA/mmBCFA pathways negatively impacts the UPR^ER, with *let-767* and *acs-1* being the most critical.

## Loss of LET-767 function impacts the UPR^ER independent of lipid depletion

Phenotypes caused by a reduction in the LET-767 enzyme could likely result from insufficient production of key metabolites, *i.e.*, mmBCFAs or LCFAs. The UPR^ER induction might then be restored to wild-type levels by supplementation of these lipids. To this end, we supplemented animals with a crude lysate composed of homogenized N2 (wild type) adult animals to provide a complete panel of lipids. We observed that supplementation of lysate was sufficient to rescue the phenotypes of *acs-1* RNAi, including suppression of the UPR^ER upon ER stress (*Figure 2—figure supplement 1A–B*). Phenotypes of *acs-1* RNAi have been shown to be a results of insufficient lipid species, suggesting that the lysate is sufficient to rescue deficiencies in lipids (*Zhang et al., 2021*). However, lysate supplementation of *let-767 RNAi* treated animals resulted in a significant improvement in organismal size, but only a slight improvement in the UPR^ER induction (*Figure 2A–B*). The ability of lysate supplementation to rescue a known essential lipid phenotype, but not suppression of the UPR^ER from let-767 RNAi, would suggest a potential mechanistic difference between the UPR^ER and size phenotypes. Although the reduced animal size is likely the result of insufficient lipids, the suppression of the UPR^ER is potentially due to other complications caused by knockdown of the *let-767* pathway.

To investigate whether lysate supplementation had an impact on ER and lipid subcellular phenotypes of *let-767* RNAi, we further characterized the impact of *let-767* RNAi and lysate supplementation on the LD and ER morphologies. We found that knockdown of *let-767* caused an extreme reduction in LD size, from large spheres to small points (*Figure 2C*). *let-767* knockdown also resulted

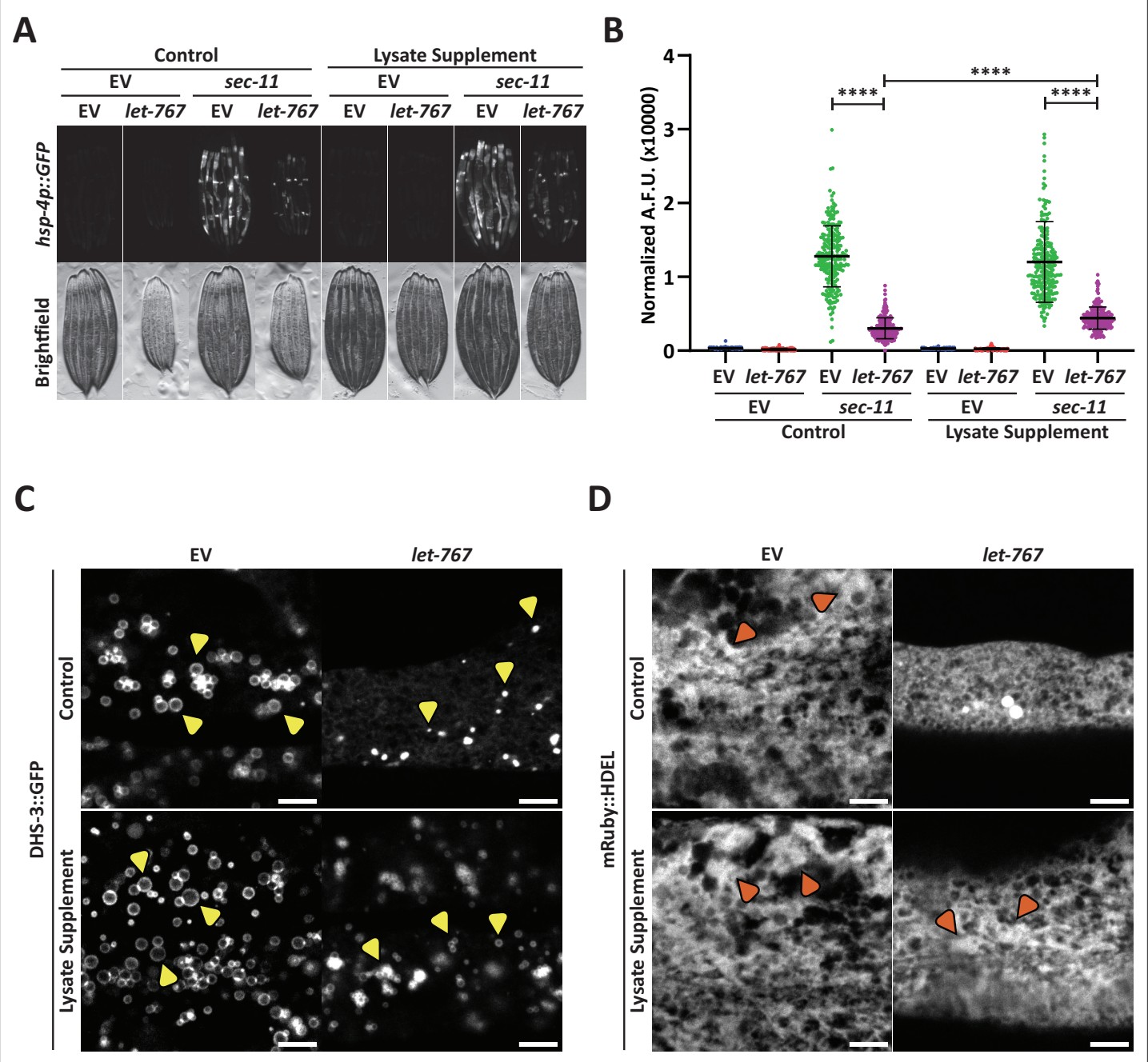

**Figure 2.** Supplementation of lysate does not restore the UPR$^{ER}$ suppressed by *let-767* RNAi. (**A**) Fluorescent micrographs of transgenic animals expressing *hsp-4p::GFP* grown on Empty Vector (EV) or *let-767* RNAi combined in a 1:1 ratio with either EV or *sec-11* RNAi supplemented with vehicle or N2 lysate to assay effects on UPR$^{ER}$ induction. (**B**) Quantification of (**A**) normalized to size using a BioSorter. Lines represent mean and standard deviation. n=189. Mann-Whitney test p-value ****<0.0001. Representative data shown is one of three biological replicates. (**C**) Representative fluorescent micrograph projections of day 1 adult transgenic animal expressing LD-localized *dhs-3::GFP*, grown on EV or *let-767* RNAi with or without N2 lysate supplementation to assay LD quality. Yellow arrowheads point to example lipid droplets. Scale bar, 5 µm. (**D**) Representative fluorescent micrograph projections of day 1 adult transgenic animal expressing ER lumen-localized *mRuby::HDEL*, grown on EV or *let-767* RNAi with or without N2 lysate supplementation to assay ER quality. Orange arrowheads point to wide ER structures. Scale bar, 5 µm. Images for organelle markers individually contrasted for clarity.

The online version of this article includes the following figure supplement(s) for figure 2:

**Figure supplement 1.** Supplementation of mmBCFA or oleic acid does not restore the UPR$^{ER}$ suppressed by *let*-RNAi.

in substantial changes to the ER morphology, practically eliminating the wider structures of the ER. However, unlike the UPR[ER] induction, supplementation of lysate improved the organelle morphology appearance by restoring the presence of wide ER structures and lipid droplets, albeit at considerably lower abundance and size (*Figure 2C–D*). These observations mark a distinction between the ability of supplementation to rescue morphological and functional phenotypes of the ER caused by *let-767* RNAi. The rescue of animal size and ER morphology phenotypes by lysate supplementation hint that they are a result of insufficient lipids, while the abilities of the ER to induce the UPR[ER] and expand lipid droplets are potentially not due to the depleted levels of lipids.

One potential reason for the ineffective rescue of the UPR[ER] induction by lysate could be insufficient levels of lipids within the lysate. We sought to determine whether excess supplementation of two lipids associated with *let-767*, mmBCFAs or LCFAs (*Entchev et al., 2008*), could rescue the UPR[ER] induction of animals with reduced *let-767*. We supplemented animals grown on *let-767* and *sec-11* RNAi with exogenous isoC17, an essential mmBCFA, or oleic acid, a LCFA known to be a significant component of the *C. elegans* fatty acid content and to increase lifespan (*Imanikia et al., 2019*; *Kniazeva et al., 2004*; *Henry et al., 2016*). mmBCFA supplementation was not sufficient to rescue induction of the UPR[ER] in *let-767* knockdown animals, showing only a slight improvement in the stress response induction (*Figure 2—figure supplement 1C–D*). However, mmBCFA supplementation was sufficient to rescue phenotypes of *acs-1* RNAi, providing evidence that our supplementation effectively delivered the essential mmBCFA and was also sufficient to rescue functional phenotypes of mmBCFA insufficiency (*Figure 2—figure supplement 1E–F*). Similar to isoC17, supplementation of oleic acid showed a very minor improvement in the ER stress response of *let-767* knockdown animals (*Figure 2—figure supplement 1G–H*). These data show that while the UPR[ER] dysfunction caused by *let-767* knockdown is likely not due to the specific loss of the essential mmBCFA or LCFA (oleic acid), isoC17 is indeed essential to the UPR[ER] in the absence of *acs-1*.

## Knockdown of lipid biosynthesis pathways restores the UPR[ER] signaling under *let-767* RNAi

Since supplementation of a WT mixture of lipids was not sufficient to recover the UPR[ER] induction, but was able to rescue ER morphology, we considered whether *let-767* knockdown could be compromising the ER membrane through unbound LET-767 partners or accumulation of upstream metabolic intermediates. A disrupted membrane could hinder ER membrane protein function for ER stress signaling as well as for lipid droplet and lipid synthesis enzymes, explaining both phenotypes and why the restored ER morphology did not coincide with restored UPR[ER] function. Indeed, a disrupted ER membrane affecting ER function was suggested as the mechanistic cause of *acs-1* phenotypes, where insufficient isoC17 disrupted the ER membrane quality and hindered lipid droplet production (*Zhang et al., 2021*). Furthermore, ER stress through *sec-11* RNAi alone caused a reduction in *let-767* transcript levels (*Figure 1—figure supplement 1B*), suggesting that a reduction of *let-767* itself is not directly suppressing the UPR[ER] induction, but possibly dependent on the levels of other factors. Instead, we hypothesized that disequilibrium within the *let-767* pathway could be the cause of membrane disruption and ultimately the loss of UPR[ER] induction. To identify a lipid that could disrupt the ER membrane, we performed untargeted complex lipidomic analysis of worms treated with *let-767* RNAi (*Supplementary file 3*, *Source data 1*). We observed a reduction in most lipids, including nearly all triglycerides, which would be expected from the reduction in lipid droplets. However, we were not able to identify a potentially accumulated lipid, with many unknown features unable to be accurately identified.

Therefore, we tested whether reducing potentially accumulated upstream metabolites of *let-767* would improve the UPR[ER] activation. Since the complete enzymatic pathway for *let-767* has yet to be definitively identified and previous works have implicated the enzyme in multiple pathways, we accomplished this by knocking down the ortholog of human SREBP, *sbp-1*, a major transcriptional regulator of numerous lipogenic enzymes and pathways (*Nomura et al., 2010*). Analysis of a published RNAseq dataset of *sbp-1* RNAi treated nematodes showed downregulation of numerous lipid synthesis genes including the fatty elongation pathway genes, such as *hpo-8*, elo-1/2/4/5/6/9, and *let-767*, the latter of which we confirmed through qPCR (*Figure 3A*; *Lee et al., 2015*). We found that animals grown on *let-767* and *sbp-1* RNAi appeared larger and healthier than on *let-767* RNAi alone. More importantly, knockdown of *sbp-1* was able to significantly improve the UPR[ER] reporter induction in *let-767*

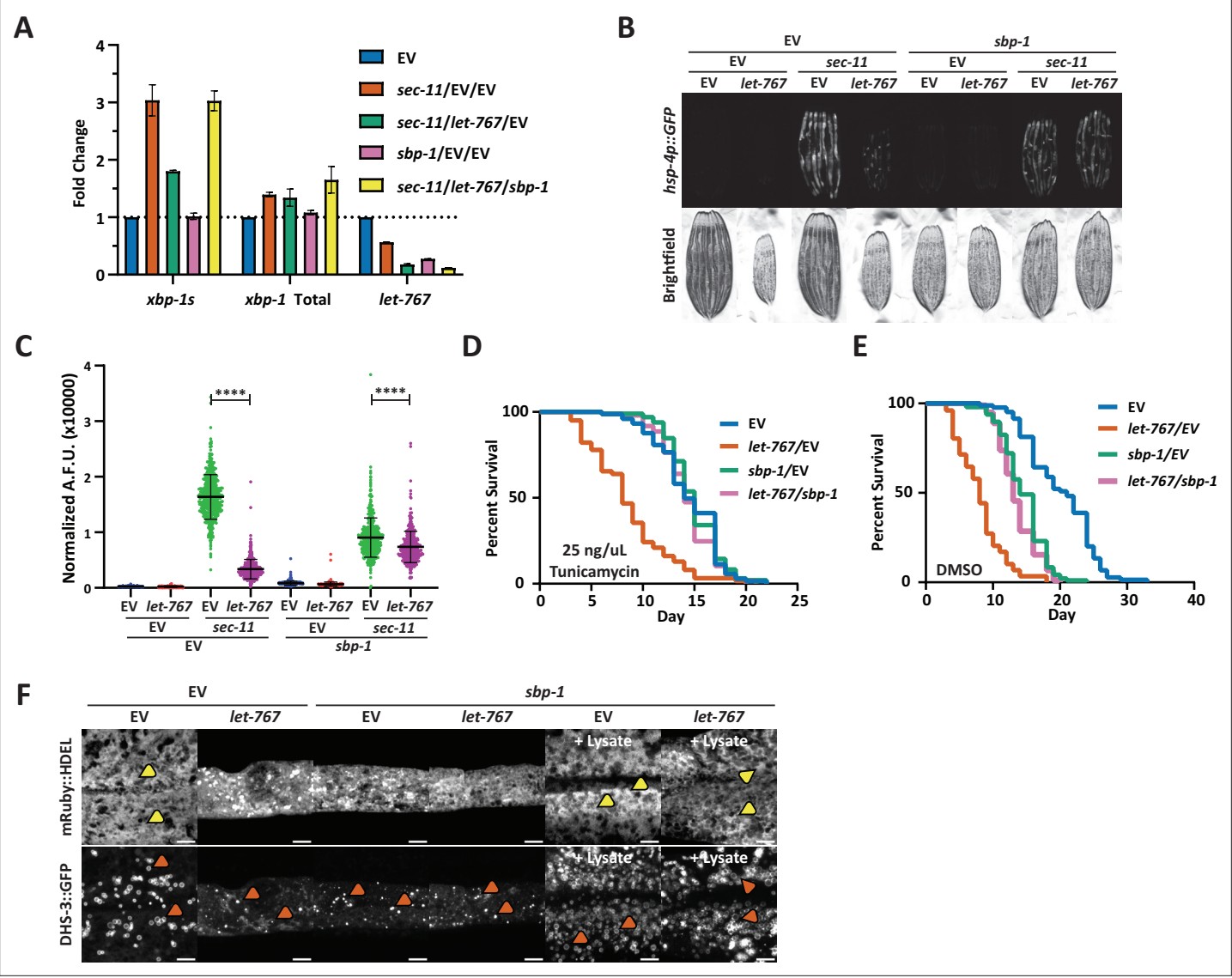

**Figure 3.** Reduced global lipid synthesis rescues the UPR<sup>ER</sup> suppression caused by *let-767* RNAi. (**A**) Quantitative RT-PCR transcript levels of *xbp-1* total, *xbp-1s*, and *let-767* from day 1 adult N2 animals grown from L1 on EV, sec-11 and EV, *let-767* and *sec-11* RNAi combined with either EV or *sbp-1* RNAi in a 1:1:1 ratio. Fold-change compared to EV treated N2 animals. Lines represent standard deviation across three biological replicates, each averaged from two technical replicates. (**B**) Fluorescent micrographs of day 1 adult transgenic animals expressing *hsp-4p::GFP* grown on EV, *let-767*, *sec-11*, and/ or *sbp-1* RNAi mixed in a 1:1:1 ratio to assay effects on the UPR<sup>ER</sup> induction. (**C**) Quantification of (**D**) normalized to size using a BioSorter. Lines represent mean and standard deviation. n=426. Mann-Whitney test p-value ****<0.0001. Representative data shown is one of three biological replicates. (**D–E**) Concurrent survival assays of N2 animals transferred to ER stress conditions of 25 ng/uL Tunicamycin (**D**) or control DMSO (**E**) conditions at day 1 of adulthood. Animals continuously grown on EV or *let-767* RNAi combined in 1:1 ratio with EV or *sbp-1* RNAi from L1 synchronization. (**F**) Representative fluorescent micrograph projections of day 1 adult transgenic animal expressing ER lumen-localized *mRuby::HDEL* or LD-localized *dhs-3::GFP*, grown on EV or *let-767* RNAi mixed in 1:1 ratio with *sbp-1* RNAi with or without N2 lysate supplementation to assay ER and LD quality. Yellow arrowheads point to wide ER structures. Orange arrowheads point to lipid droplets. Scale bar, 5 μm. Images for organelle markers individually contrasted for clarity.

knockdown animals under ER stress from *sec-11* RNAi and effectively restore splicing of *xbp-1* to *xbp-1s* (*Figure 3A–C*). To test whether the improved UPR<sup>ER</sup> signaling was indeed a functional restoration of the ER stress response, we performed an ER stress survival assay. As expected, animals treated with *let-767* RNAi have a severe defect in survival when exposed to the proteotoxic stress of Tunicamycin, an inhibitor of N-linked glycosylation. Correlating with our observations of the UPR<sup>ER</sup> reporter, *sbp-1* knockdown rescued the survival of *let-767* RNAi treated animals to the level of *sbp-1* RNAi alone (*Figure 3D–E*, *Supplementary file 2*). Interestingly, *sbp-1* RNAi itself reduced the induction level of

the UPR[ER] reporter and the survival rate of animals on Tunicamycin, suggesting that while a reduction in the entire *let-767* pathway could indeed rescue the UPR[ER] suppression caused by *let-767* RNAi, a global reduction in lipid synthesis enzymes limits the maximal induction of the UPR[ER].

Finally, we determined whether *sbp-1* RNAi would also rescue the defects in LD and ER morphology caused by *let-767* knockdown. Congruent with SREBP's central role in promoting lipogenesis, we observed a depletion of lipid droplets and a slight perturbation of the ER morphology with *sbp-1* RNAi (*Figure 3F*). In combination with *let-767* RNAi, *sbp-1* knockdown did not rescue the ER and lipid droplet morphology to WT conditions, highlighting that restoring the UPR[ER] signaling is not completely dependent on restoring ER lipid levels and morphology. However, in the presence of lysate supplementation, *sbp-1* RNAi alone or combined with *let-767* RNAi resulted in ER and lipid droplet morphology resembling that of WT animals with the presence of wide sheet-like structures and larger lipid droplets. Therefore, a more complete rescue of the phenotypes exhibited by *let-767* knockdown animals could be achieved by a combination of (1) the reduction of the *let-767* pathway through knockdown of *sbp-1* to reduce global lipid synthesis, and (2) exogenous supplementation of the lipids to restore lipids lost by *sbp-1* and *let-767* RNAi.

## *let-767* knockdown impacts *xbp-1* splicing and *xbp-1s* activity

Proper UPR[ER] signaling is dependent on dimerization of IRE-1 at the ER membrane to splice *xbp-1* mRNA to its active *xbp-1s* isoform. Therefore, one possible mechanism by which loss of *let-767* can impact the UPR[ER] is by impeding IRE-1 activity through altered membrane dynamics. Indeed, knockdown of *let-767* results in altered ER membrane dynamics where the mobile fraction of an ER transmembrane protein (SPCS-1) is significantly reduced when measured by Fluorescence Recovery After Photobleaching (FRAP) (*Figure 4A–B*). The percent mobile being the calculated fraction of fluorophore that can move within the bleached area during the FRAP process. Supplementation of lysate improved the ER membrane mobility to 62% of WT levels. Comparatively, ER luminal mRuby dynamics, which would likely be indirectly affected by the global changes in ER 3D structure, were less severely impacted by *let-767* RNAi. These effects were rescued to a greater extent by lysate supplementation, potentially due to the recovered ER structure, restoring the mobile fraction of *let-767* RNAi treated animals to 84% of WT levels (*Figure 4C–D*). While specific changes in dynamics are likely dependent on the individual protein being observed, our results provide evidence that membrane dynamics have been altered when *let-767* is knocked down.

To determine whether the UPR[ER] was indeed being affected at the ER membrane, we examined the impact of *let-767* knockdown on ectopic UPR[ER] activation at two different points along the mechanistic pathway: (1) overexpression of *ire-1*, which would ectopically activate the UPR[ER] at the membrane by constitutively splicing *xbp-1* and (2) overexpression of the active *xbp-1s* isoform, which would bypass the requirement of IRE-1 splicing at the ER membrane (*Li et al., 2010*). In creating the necessary strains, we found that intestinal over-expression of the full-length *ire-1a* isoform proved to be lethal. However, overexpression of the *ire-1b* isoform lacking the luminal domain, fused with the mRuby fluorophore (*mRuby::ire-1b*), was viable and had constitutive activation of the UPR[ER]. To ensure that our fusion protein was responsible for activating the UPR[ER], we knocked down *ire-1* through an RNAi targeting the N-terminal region only found in *ire-1a*. The *ire-1a* RNAi was able to suppress the UPR[ER] signaling in WT animals with and without ER stress but did not eliminate the basal UPR[ER] signal in animals expressing *mRuby::ire-1b*. Conversely, RNAi targeting common sequences of both *ire-1a* and *ire-1b* suppressed the UPR[ER] induction in both WT and animals expressing our *mRuby::ire-1b* construct, even when ER stress was induced by *sec-11* RNAi. (*Figure 5—figure supplement 1A–D*). Therefore, our *mRuby::ire-1b* construct was sufficient to induce the intestinal UPR[ER] without the endogenous *ire-1a*. Likewise, we confirmed that overexpression of our intestinal *mRuby::xbp-1s* construct was able to induce the UPR[ER] in an *xbp-1 (zc12)* null-mutant background (*Figure 5—figure supplement 1E–F*). Beyond observing that our construct was sufficient to induce the UPR[ER], we also found that loss of endogenous *xbp-1* resulted in an even higher level of basal UPR[ER] induction, potentially due to a negative regulatory role for unspliced *xbp-1*, that has been previously suggested (*Yoshida et al., 2006*).

Next, we tested UPR[ER] induction of our *ire-1* and *xbp-1s* overexpression strains on *let-767* RNAi. *let-767* knockdown reduced the induction of the UPR[ER] reporter in *mRuby::ire-1b* animals (*Figure 5A–B*). However, *let-767* RNAi also reduced the UPR[ER] reporter induction in the *mRuby::xbp-1s, xbp-1(zc12)*

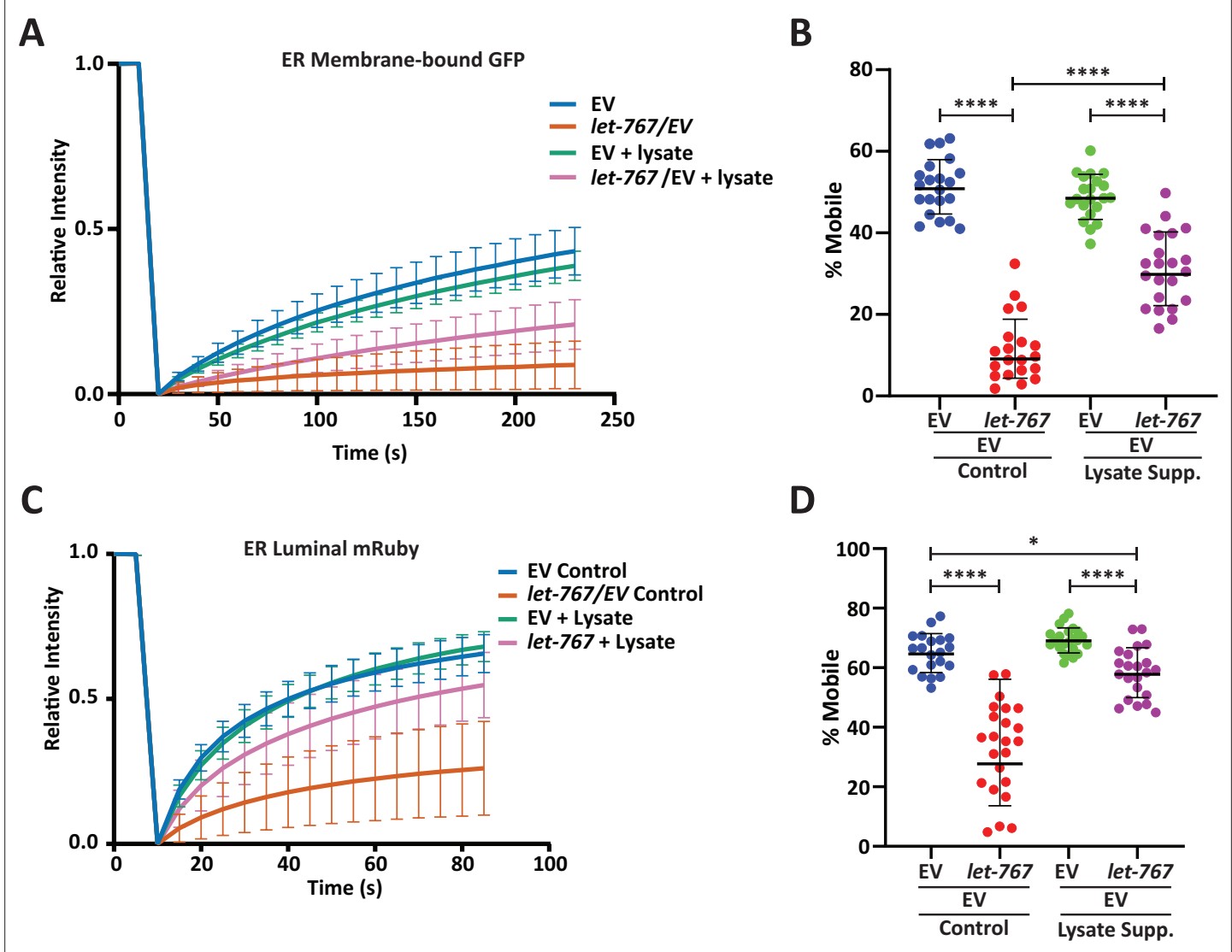

**Figure 4.** Lysate supplementation rescues luminal ER protein dynamics but not ER membrane protein dynamics. (**A**) FRAP curve of intestinal ER-transmembrane protein, SPCS-1::GFP, from day 1 adult animals grown on Empty Vector (EV) or *let-767* RNAi mixed with EV in a 1:1 ratio supplemented with vehicle or N2 lysate (geometric mean for n=20 pooled from two biological replicates). Lines represent geometric standard deviation. (**B**) Calculated percent mobile SPCS-1::GFP of (**A**). Lines represent geometric mean and geometric standard deviation. Mann-Whitney test p-value ****<0.0001. (**C**) FRAP curve of intestinal ER-lumen protein, mRuby::HDEL, from day 1 adult animals grown on EV or *let-767* RNAi mixed with EV in a 1:1 ration supplemented with vehicle or N2 lysate (geometric mean for n=20 pooled from two biological replicates). Lines represent geometric standard deviation. (**D**) Calculated percent mobile mRuby::HDEL of (**A**). Lines represent geometric mean and geometric standard deviation. Mann-Whitney test p-value ****<0.0001 and *<0.05.

strain (***Figure 5C–D***). By utilizing the *mRuby::xbp-1s, xbp-1 (zc12)* null-mutant we eliminated the possibility that the reduced induction was due to the negative regulatory effects of unspliced *xbp-1s*. To ensure that the reduced UPR<sup>ER</sup> induction was not due to altered expression of our *mRuby::xbp-1s* construct, we performed qPCR for *xbp-1* in animals grown on *let-767* RNAi. We observed a significant drop in the percent of *xbp-1* splicing in animals expressing *mRuby::ire-1b* when they were grown on *let-767* RNAi, but not in animals expressing *mRuby::xbp-1s* (***Figure 5E–F***). However, the reduced UPR<sup>ER</sup> induction despite the similar levels of *xbp-1s* transcripts between EV and *let-767* RNAi suggests that in addition to affecting *xbp-1* splicing, *let-767* RNAi is likely affecting *xbp-1s* activity downstream of its splicing.

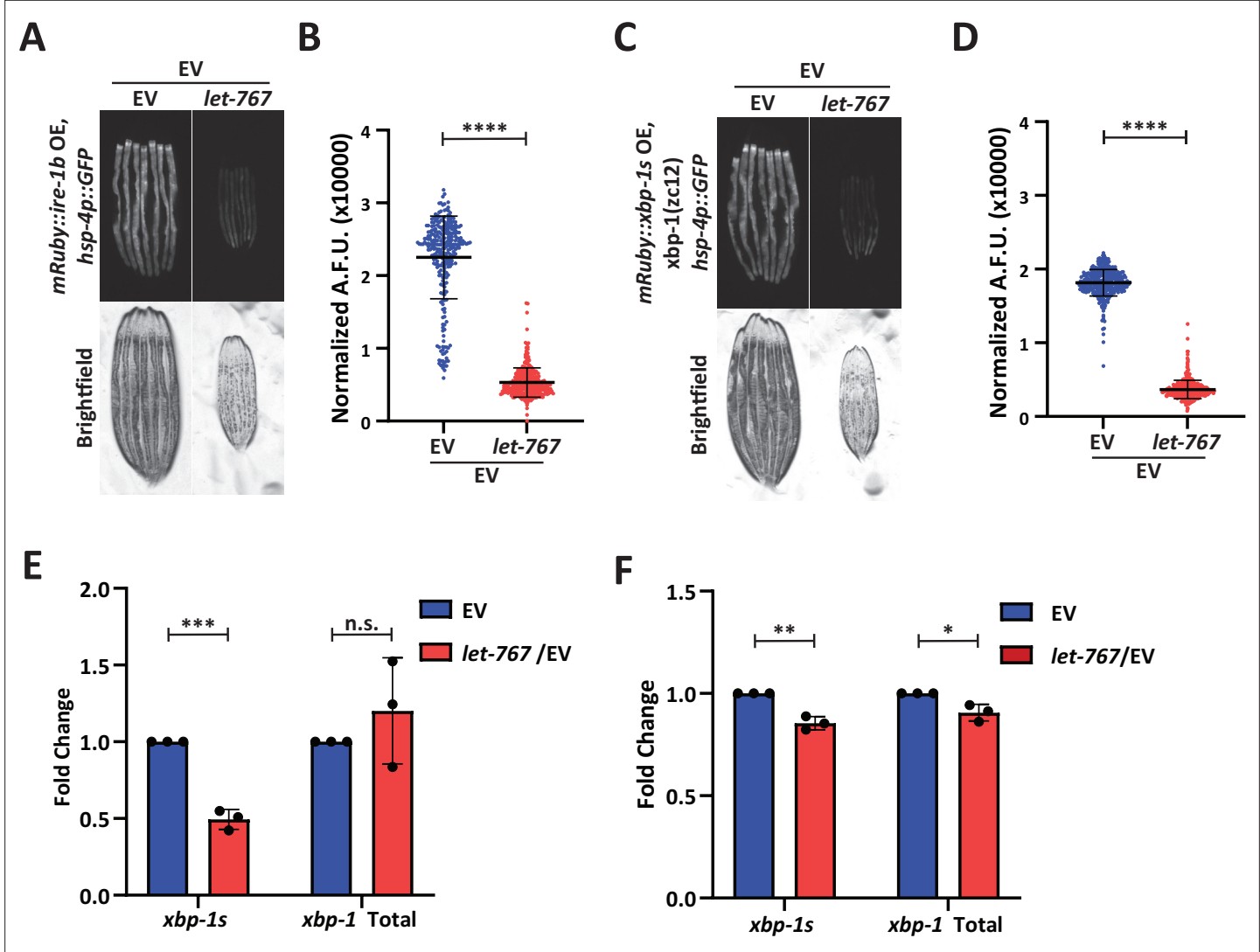

**Figure 5.** *let-767* knockdown impacts UPR<sup>ER</sup> induction independent of *xbp-1* splicing. (**A**) Fluorescent micrographs of day 1 adult transgenic animals expressing *hsp-4p::GFP* and intestinal *mRuby::ire-1b* grown on Empty Vector (EV) or *let-767* RNAi mixed with EV in a 1:1 ratio to assay the UPR<sup>ER</sup> induction. (**B**) Quantification of (**A**) normalized to size using a BioSorter. Lines represent mean and standard deviation. n=290. Mann-Whitney test P-value ****<0.0001. Representative data shown is 1 of 3 biological replicates. (**C**) Fluorescent micrographs of day 1 adult *xbp-1(zc12)* transgenic animals expressing *hsp-4p::GFP* and intestinal *mRuby::xbp-1s* grown on EV or *let-767* RNAi mixed with EV in a 1:1 ratio to assay the UPR<sup>ER</sup> induction. (**D**) Quantification of (**C**) normalized to size using a BioSorter. Lines represent mean and standard deviation. n=398. Mann-Whitney test p-value ****<0.0001. Representative data shown is one of three biological replicates. (**E**) Quantitative RT-PCR transcript levels of *xbp-1s* and total *xbp-1* from day 1 adult *mRuby::ire-1b* animals grown from L1 on *let-767* RNAi mixed 1:1 with EV. Fold-change compared to EV treated animals.Unpaired t-test p-value ***<0.0005. Error bars indicate ± standard deviation across three biological replicates, each averaged from two technical replicates. (**F**) Quantitative RT-PCR transcript levels of *xbp-1s* and total *xbp-1* from day 1 adult *mRuby::xbp-1s* animals grown from L1 on *let-767* RNAi mixed 1:1 with EV. Fold-change compared to EV treated animals. Unpaired t-test p-value **<0.005 and *<0.05. Error bars indicate ± standard deviation across three biological replicates, each averaged from two technical replicates. Dots indicate averaged biological replicate values.

The online version of this article includes the following figure supplement(s) for figure 5:

**Figure supplement 1.** Overexpression of *ire-1b* or *xbp-1s* induces the UPR<sup>ER</sup> independently of endogenous *ire-1a* or *xbp-1*, respectively.

## The *let-767* upstream metabolite, 3-oxostearic acid, reduces induction of the UPR<sup>ER</sup>

To identify a node within the *let-767* pathway that might be responsible for the disruptive metabolite or protein, we utilized mammalian cell culture to probe the effect of specific metabolites in the *let-767* pathway on ER homeostasis. This would allow us to saturate cellular exposure to lipid intermediates

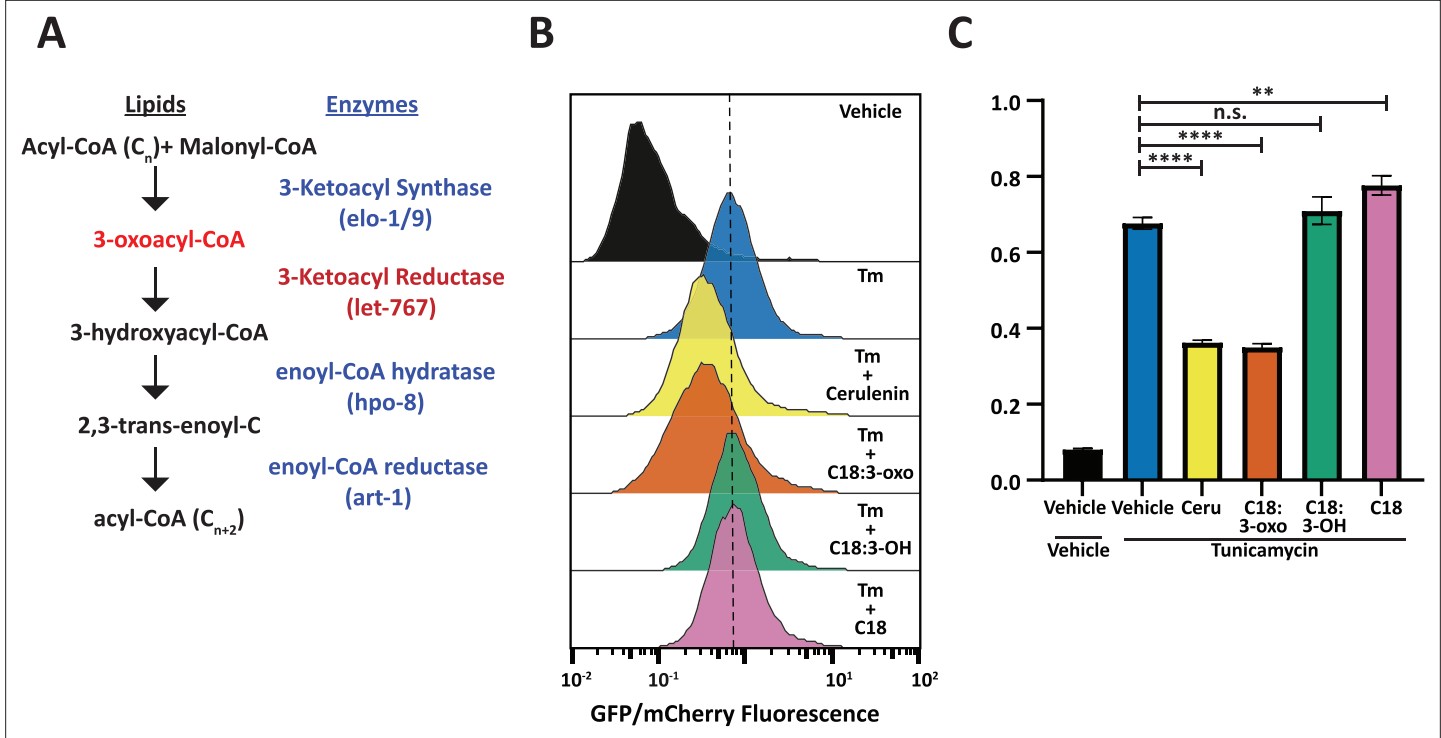

**Figure 6.** Fatty acid intermediate, 3-oxoacyl, is sufficient to reduce UPR$^{ER}$ induction. (**A**) Fatty acid elongation pathway displaying intermediate lipid metabolites and annotated *C. elegans* genes. (**B**) Flow cytometry measurement histogram of huh7 UPRE reporter fluorescence normalized to EIF2A promoter driving mCherry. Cells were treated for 18 hr with Tunicamycin and vehicle, 50 μM Cerulenin, 50 μM 3-oxostearic acid, 50 μM stearic acid, or 50 μM 3-hydroxystearic acid. Data is representative of three biological replicates. (**C**) Median bar graph of (**B**). Unpaired t-test p-value **=0.004 and ****<0.0001. Error bars indicate ± standard deviation across three technical replicates.

and avoid the possibility of bacteria processing the intermediates prior to digestion by *C. elegans* animals. Previous works have provided evidence that the human ortholog of *let-767* are HSD17B12 and HSD17B3 (*Entchev et al., 2008*). Interestingly, LET-767 shares the most protein sequence percent identity (39.87%) with HSD17B12, the 3-ketoacyl reductase, which caused similar phenotypes in mice when knocked out. HSD17B12 was found to be essential for development in mice and knockout of the gene in adult mice resulted in reduced body weight, reduced lipid content, and caused liver toxicity hypothesized to be from accumulation of toxic intermediates (*Rantakari et al., 2010*; *Heikelä et al., 2020*).

3-Ketoacyl reductases perform the second step in fatty acid synthesis/elongation, metabolizing 3-oxoacyl-CoA to 3-hydroxyacyl-CoA (*Figure 6A*; *Moon and Horton, 2003*). While the fatty acid elongation pathway has not been shown to affect the UPR$^{ER}$, use of the fatty acid synthase inhibitor, Cerulenin, has been shown to increase levels of XBP1s with reduced transcriptional activity due to changes in palmitoylation (*Chen et al., 2020*). From these studies, we hypothesized that the accumulation of the metabolites upstream of LET-767 could be reducing the transcriptional activity of XBP-1S. To this end, we tested whether an upstream metabolite of LET-767 was sufficient to reduce UPR$^{ER}$ induction in huh7 cells containing a 5 x unfolded protein response element (UPRE) GFP reporter and EIF2A promoter driving mCherry. As fatty acid elongation extends fatty acids beyond the 16 carbons synthesized by fatty acid synthase, we supplemented our reporter line with the 18 carbon fatty acid metabolites upstream and downstream of the 3-ketoacyl reductase in combination with Tunicamycin to induce ER stress (*Figure 6B–C*). We observed that the upstream metabolite, 3-oxostearic acid, reduced the normalized 5xUPRE reporter induction. In comparison, the downstream metabolites, 3-hydroxystearic acid and stearic acid, did not reduce the normalized 5xUPRE reporter signal. Instead, stearic acid caused a slight induction of the reporter signal, in agreement with studies showing that saturated lipids induce the UPR$^{ER}$ (*Volmer et al., 2013*). This indicated that the 5xUPRE was specifically

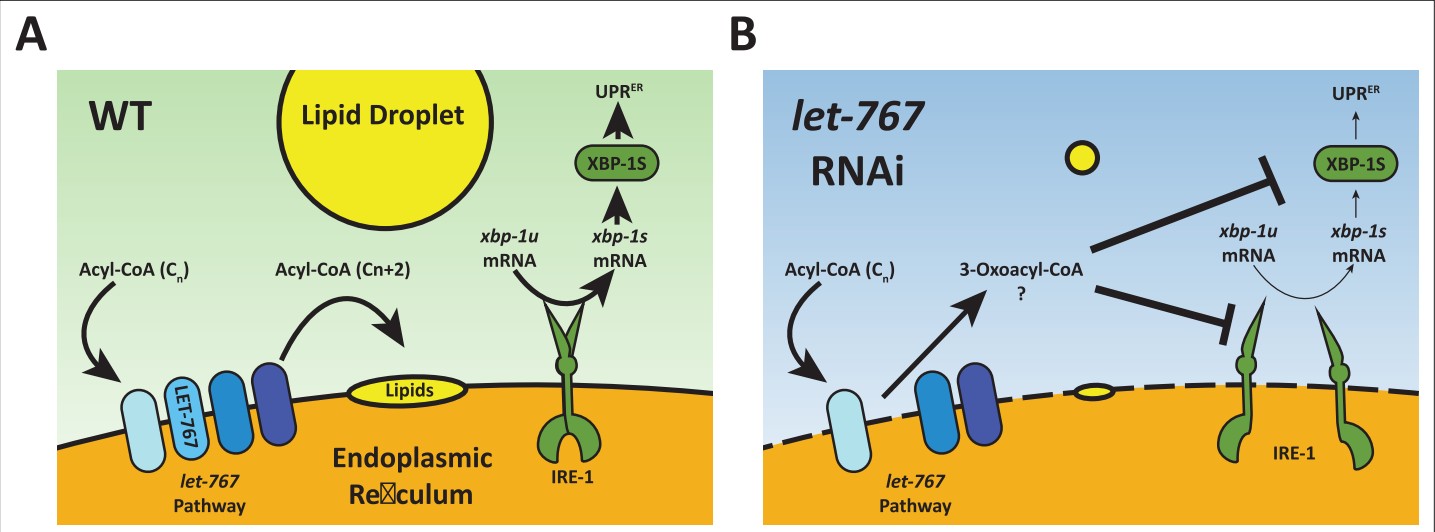

**Figure 7.** Model for *let-767* RNAi blocking UPR[ER] induction. (**A**) Under WT condition, acyl-CoA metabolites are elongated by the *let-767*/HSD17B12 pathway and utilized at the membrane to synthesize other lipids such as neutral lipids stored in lipid droplets. IRE-1 responds to ER stress by splicing *xbp-1u* to *xbp-1s*, which is then translated and able to induce expression of the UPR[ER] target genes (**B**) Knockdown of *let-767* results in disequilibrium of *let-767*/HS17B12 pathway, leading to accumulation of intermediates such as 3-oxoacyl-CoA and reduced lipid production. Intermediate metabolites disrupt membrane quality (dashed line) and negatively affect induction of the UPR[ER] by reducing splicing of *xbp-1s* (smaller arrows) by IRE-1 and reducing the function of *xbp-1s* post-splicing (smaller arrows).

sensitive to the 3-oxo metabolite upstream of the *let-767* pathway and that this interaction between lipid metabolism and the UPR[ER] is likely conserved in mammalian cells.

## Discussion

Cells must monitor and maintain both lipid and protein homeostasis to preserve cellular function. The ER is uniquely a major site of both protein and lipid synthesis. Here, we performed a genetic screen to identify lipid related genes that impact the ER response to protein stress and found *let-767* to be necessary for proper animal size, organelle morphology, neutral lipid accumulation, and induction of the UPR[ER]. Under WT conditions, the *let-767*/HSD17B12 pathway elongates fatty acids that are then used to produce other lipids, such as triglycerides stored in lipid droplets (**Figure 7A**). The UPR[ER] is also able to respond to ER stress by inducing XBP-1s activity. When the *let-767* is knocked down, the pathway is unable to elongate fatty acids and accumulates the upstream metabolite, 3-oxoacyl-CoA, and possibly other metabolic intermediates that disrupt the ER membrane quality (**Figure 7B**). Cells are unable to synthesize downstream lipids for lipid droplets and either through disruption of the membrane or direct interaction with 3-oxoacyl-CoA, *xbp-1* splicing and the activity of spliced *xbp-1s* is significantly reduced in the presence of protein induced ER stress. While unfolded proteins and lipid disequilibrium are known to independently induce the UPR[ER], our novel finding demonstrates that changes in lipid pathways could significantly impact a cell's ability to respond to protein stress and brings to light a potential mechanism through which lipid disequilibrium might facilitate the progression of proteopathic diseases.

LET-767 has been characterized as a hydroxysteroid dehydrogenase with evidence as both a steroid modifying enzyme and a 3-ketoacyl reductase located on the ER, consistent with mammalian HSD17B3 and HSD17B12 (*Entchev et al., 2008*; *Desnoyers et al., 2007*). LET-767 has also been implicated as a requirement for branched-chain and long-chain fatty acid production, however, its linear metabolic pathway has not been thoroughly investigated so its exact lipid products or protein interactors are yet to be identified. Through our depletion of mmBCFA and LCFA pathway genes, we find that both pathways are essential to having a maximal ER unfolded protein response to protein stress, with *acs-1* RNAi having the most similar phenotypes to *let-767* knockdown. While individual supplementation of mmBCFAs or LCFAs did not rescue the *let-767* RNAi phenotypes, supplementation of a more complex lipid mixture, crude worm lysate, was able to significantly rescue organelle

morphology, size, and reproduction phenotypes. This suggests that loss of *let-767* has a more global effect on lipid production rather than affecting a single lipid species. A straightforward explanation for its requirement in global lipid equilibrium and the UPR^ER could be that the *let-767* pathway is crucial for an elemental component of the ER, such as production of fundamental lipids required for ER membrane quality as has been shown for the *acs-1* pathway (*Zhang et al., 2021*). The compromised ER integrity caused by *acs-1* RNAi demonstrates that a in addition to affecting lipid droplet production, a disrupted ER membrane can impact UPR^ER induction and that these phenotypes can be rescued through correction of lipid deficiencies such as lipid supplementation.

We would then expect that *let-767* levels would be critical for ER functions, including UPR^ER signaling. However, the inability of supplementation to rescue *let-767* RNAi phenotypes would suggest a different mechanism for the compromised ER membrane integrity. Interestingly, ER stress results in reduced *let-767* transcript levels. Furthermore, the reduced UPR^ER signaling caused by *let-767* RNAi was significantly rescued by knockdown of the upstream transcription factor, *sbp-1*, which also reduced *let-767* transcript levels and lipid stores. A possible interpretation of these results is that an alternative pathway to produce the essential lipids is upregulated under ER stress or *sbp-1* RNAi, but with the existence of an alternative pathway, the UPR^ER would not be suppressed by knockdown of *let-767*. Instead, we propose that the *let-767* RNAi phenotypes are caused by disequilibrium of the fatty elongation pathway at the *let-767* node. By reducing this specific node of the fatty elongation pathway, intermediate metabolites upstream of *let-767* could interfere with ER membrane dynamics, interactions, and functions. A disrupted ER membrane would also explain the similarities in phenotypes with *acs-1* RNAi, which has been shown to affect ER integrity and which we show here can impact UPR^ER induction (*Zhang et al., 2021*). As the UPR^ER sensors and numerous lipogenic enzymes reside on the ER membrane, a disrupted ER membrane would explain why knockdown of *let-767* and other lipid genes affects the capacity of the UPR^ER induction. While knockdown of *sbp-1* also reduced the level of UPR^ER induction, the reduction of numerous lipogenic pathways, including the entire *let-767* pathway, was able to improve the UPR^ER function of animals on *let-767* RNAi compared to *let-767* RNAi alone. Mechanistically, this could function by preventing the lipid disequilibrium caused by intermediates of the fatty acid elongation pathway by reducing the entire pathway instead of just one node. While our untargeted lipidomic analysis did not reveal an increase in any specific lipid, it did confirm that *let-767* RNAi disrupts lipid levels globally. With multiple types of fatty acids of different lengths being elongated by the same pathway, identifying a single lipid species may prove difficult since numerous 3-oxoacyls may be contributing to the change in ER membrane quality.

To test whether the ER membrane disorganization was the source of UPR^ER dysfunction, we aimed to bypass the splicing of *xbp-1* by IRE-1 at the ER membrane by overexpressing the already spliced isoform, *xbp-1s*. However, *let-767* RNAi still caused a reduction in the ER stress response. The reduced UPR^ER induction in animals overexpressing *xbp-1s* points to an additional mechanism downstream of splicing for the muted UPR^ER. The regulation of XBP1 mRNA has been proven to be more complex than simple splicing upon ER stress. XBP1u mRNA is required to localize to the ER membrane to facilitate splicing to XBP1s upon ER stress through a hydrophobic region and translational stalling (*Gómez-Puerta et al., 2022*; *Yanagitani et al., 2009*; *Yanagitani et al., 2011*). Without proper localization, *xbp-1u* would be translated and negatively regulate any *xbp-1s* (*Yoshida et al., 2006*). Additionally, there are potentially other factors residing on the membrane that are required for effective translation of *xbp-1s*, such as specific ribosomal complexes or mRNA stabilizing factors. An altered ER membrane would impact every interaction of *xbp-1* mRNA with the ER membrane, including transient interactions that have yet to be discovered.

While splicing is a major point of regulation for *xbp-1*, previous studies have found other potential nodes of regulation for *xbp-1* through SUMOylation, deacetylation, and palmitoylation (*Chen et al., 2020*; *Bang et al., 2019*; *Jiang et al., 2012*). Disruption of the ER membrane by *let-767* RNAi could impact these other pathways affecting fatty acid synthase activity or protein palmitoylation. Through supplementation of fatty acid elongation intermediates to human huh7 hepatocytes, we provide evidence that increased levels of the 3-oxoacyl metabolite that is upstream of the *let-767* ortholog, HSD17B12, a 3-ketoacyl reductase, is sufficient to reduce the ER stress response to a similar level as the fatty acid synthase inhibitor, Cerulenin. Mechanistically, Cerulenin has been proposed to prevent palmitoylation of XBP1S which reduced its transcriptional activity without impacting its protein levels

(*Chen et al., 2020*). Whether the 3-ketoacyl metabolite impacts fatty acid synthase activity or protein palmitoylation requires further investigation.

Fatty acids exist in numerous configurations of varying lengths and combination of branches and double bonds. These lipids are one of the basic building blocks that compose the membranes of the cell and their qualities can have drastic effects on membrane thickness, fluidity, and curvature (*Harayama and Riezman, 2018*). Therefore, it is not surprising that alterations in lipids such as mmBCFA levels can impact ER function (*Zhang et al., 2021*). Our work demonstrates that incomplete processing of lipids can also impact ER function, in our case by knock down a key enzyme within the fatty elongation pathway. Further work into how different nodes within lipid pathways and how levels of lipid intermediates change with age would greatly contribute to growing amount research on the relationship between lipid changes and aging, including age-dependent neurological diseases (*Gille et al., 2021*; *Yoon et al., 2022*). However, this work may prove to be labor intensive due to the complexity of lipidomics and the possible disconnect between transcript levels, protein levels, and enzymatic activity with age.

## Materials and methods

### Nematode strains

N2 Bristol, LIU1 (ldrIs[dhs-3p::dhs-3::GFP]), SJ4005 (zcIs4[hsp-4p::GFP]), SJ17 (xbp-1(zc12) III; zcIs4 [hsp-4p:GFP] V), SJ4100 (zcIs13[hsp-6p::GFP]), CL2070 (dvIs70[hsp-16.2p::GFP]), VS25 (hjIs[vha-6p::GFP::C34B2.10(SP12) +unc-119(+)]),EG6703 (unc-119(ed3); cxTi10816; oxEx1582[eft-3p::GFP +Cbr-unc-119]) strains were obtained from the *Caenorhabditis* Genetics Center (CGC). AGD2192 (uthSi60[vha-6p::ER-signal-sequence::mRuby::HDEL::unc-54 UTR, cb-unc-119(+)] I; unc-119(ed3) III) (*Daniele et al., 2020*). Transgenic strains created for this study were generated from EG6703 via the MosSCI method (*Yanagitani et al., 2011*) or through crossing strains.

Transgenic strains created:

> AGD2424 (unc-119(ed3) III; uthSi65[vha-6p::ERss::mRuby::ire-1a (344-967aa)::unc-54 3'UTR cb-unc-119(+)] IV)
> AGD2425 (uthSi65[vha-6p::ERss::mRuby::ire-1a (344-967aa)::unc-54 3'UTR cb-unc-119(+)] IV; zcIs4[hsp-4p::GFP] V)
> AGD2012 (unc-119(ed3) III; uthSi71[vha-6p::mRuby::xbp-1s::unc-54 3'UTR cb-unc-119(+)] IV)
> AGD2735 (uthSi71[vha-6p::mRuby::xbp-1s::unc-54 3'UTR cb-unc-119(+)] IV; zcIs4[hsp-4p::GFP] V)
> AGD2996 (xbp-1(zc12) III; uthSi71[vha-6p::mRuby::xbp-1s::unc-54 3'UTR cb-unc-119(+)] IV; zcIs4[hsp-4p::GFP] V)

### Worm growth and maintenance

All worms were maintained at 20 °C on NGM agar plates seeded with OP50 *E. coli* bacteria. Prior to experiments, worms were bleach synchronized as described in *Higuchi-Sanabria et al., 2018*, followed by overnight L1 arrest in M9 buffer (22 mM KH2PO4 monobasic, 42.3 mM Na2HPO4, 85.6 mM NaCl, 1 mM MgSO4) at 20 °C. For RNAi experiments, arrested L1s were plated on 1 µm IPTG, 100 µg/mL Carbenicillin NGM agar plates maintained at 20 °C and seeded with RNAi bacteria grown in LB +100 µg/mL Carbenicillin.

### Fluorescent microscopy

Transcriptional reporter strains were imaged using a Leica DFC3000 G camera mounted on a Leica M205 FA microscope. Worms were grown to day 1 of adulthood at 20 °C, hand-picked, and immobilized with 100 mM Sodium Azide M9 buffer on NGM agar plates. Raw images were cropped, and contrast matched using ImageJ software.

For the initial screen of hsp-4p::GFP reporter animals, fluorescence was scored by eye using the following criteria: 2=increased fluorescence, 1=possible increase in fluorescence, 0=no change, –0.5=small regions of dimmer fluorescence, –1=small regions of complete loss of fluorescence, –1.5=globally dimmer fluorescence and some regions of no fluorescence, –2=global loss

of fluorescence and small regions of dim fluorescence, –2.5=global loss of fluorescence except for regions within spermatheca, –3=complete loss of fluorescence.

Confocal images were acquired using a Leica Stellaris 5 confocal platform with a ×63 objective. Day 1 worms were picked onto 100 mM Sodium Azide M9 buffer on slides and imaged within 30 min. Raw images were cropped and independently contrast optimized for clarity using ImageJ software.

### Biosorter analysis

Transcriptional reporter strains were grown to day 1 of adulthood at 20 °C. Animals were collected into a 15 mL conical tube with M9 buffer and allowed to settle at the bottom before supernatant was aspirated. Animals were resuspended in M9 buffer and analyzed using a Union Biometrica COPAS Biosorter (P/N: 350-5000-000) as described in *Bar-Ziv et al., 2020*. Animals which saturated the signal capacity of 65532 or were outliers in both animal size parameters (*i.e.*, Extinction and Time Of Flight) were censored. Mann-Whitney statistical tests were performed on fluorescence normalized to animal extinction using GraphPad Prism software. Integrated fluorescence is normalized to integrated extinction (as a proxy for size) during quantification. Biosorter plots shown are of populations from a single experiment.

### qPCR

Animals grown on EV or RNAi bacteria were collected at day 1 of adulthood using M9 and washed 3 x. M9 was aspirated and trizol added before 3 cycles of freeze/thaw in liquid nitrogen. Chloroform was then added at a ratio of 1:5 (chloroform:trizol). Separation of RNA was performed through centrifugation in gel phase-lock tubes. The RNA aqueous phase was transferred into new tubes containing isopropanol. RNAi was purified using the QIAGEN RNeasy Mini Kit (74106) according to the manufacturer's instructions. cDNA was synthesized using 2 µg of RNA and the QIAGEN QuantiTect Reverse Transcriptase kit (205314) according to the manufacturer's instructions. qPCR was performed using SYBR-green. Analysis was performed for each biological replicate using the Delta-Delta CT method with *pmp-3*, *cdc-42*, and *Y45F10D.4* as housekeeping genes (*Hoogewijs et al., 2008*).

### Lysate supplementation

Crude lysate was obtained from N2 worms grown on 40 x concentrated EV bacteria at 20 °C.~120,000 day 1 adult worms were collected with M9 and washed 6 x with M9 buffer. Animals were then homogenized 20 x in 3 mL of M9 buffer using an ice-cold 15 mL Dura-Grind Stainless Steel Dounce Tissue Grinder (VWR, 62400–686). Crude lysate was transferred to 1.5 mL tubes and frozen in liquid N2.

Supplementation experiments were prepared by mixing 4 x concentrated RNAi bacteria with crude lysate at a 2:1 ratio, respectively. The lysate mixture was plated on RNAi plates and allowed to dry. Dried plates were then UV irradiated without lids for 9 min in an ultraviolet crosslinker (UVP, CL-1000) before plating L1 arrested worms on the plate.

### Lipid supplementation

Lipid supplementation experiments were prepared by inoculating cultures (LB +100 µg/mL Carbenicillin) with RNAi bacteria or empty vector and then adding the lipids to the specified concentration or equal volumes of ethanol. The cultures were allowed to grow overnight to saturation and then concentrated to 4 x before being plated on RNAi plates. The bacteria was allowed to dry overnight and then UV irradiated without lids for 9 min in an ultraviolet crosslinker (UVP, CL-1000) before plating L1 arrested worms on the plate.

### Tunicamycin survival assay

Tunicamycin survival assays were conducted on NGM agar plates containing 25 ng/µL of tunicamycin in DMSO, or equal volume of DMSO with specified RNAi bacteria at 20 °C. Animals were moved daily for 4–7 days to new RNAi plates until progeny were no longer observed. Worms with protruding intestines, bagging phenotypes, or other forms of injury were scored as censored and not counted as part of the analysis. For combined RNAi lifespans, saturated cultures were mix 1:1 by volume.

## FRAP analysis

FRAP 4D images were acquired using a 3i Marianas spinning-disc confocal platform. Photobleaching of a 10 μm x 10 μm region within a 4 μm Z-stack was performed using a 488 nm laser for 1–2ms. Raw images were processed with FIJI (*Schindelin et al., 2012*) into sum Z-projections and aligned using the 'RigidBody' setting of the StackReg ImageJ plugin (*StackReg, 2021*). FRAP analysis was then performed with the FRAP Profiler ImageJ plugin (*FRAP profiler plugin, 2021*) by selecting the 10 μm x 10 μm photobleached region as region 1 and the entire fluorescent area as region 2. Briefly, the plugin calculates percent mobile by fitting an exponential curve to the plotted normalized recovered fluorescence values to define the recovery curve and determine the fraction/percent that is able to move within the ROI.

## Lipidomic analysis

N2 animals were grown on 40 x concentrated EV or *let-767* RNAi bacteria at 20 °C, respectively. Animals were collected with M9 and frozen in liquid nitrogen to be submitted to the UC Davis West Coast Metabolomics Center (WCMC) for untargeted complex lipid analysis. Peak values of lipids identified by the WCMC were first corrected by subtracting corresponding peaks of blanks and then their abundance estimated relative to internal standards of their corresponding lipid class before calculating average fold change from 6 biological replicates.

## Cell Culture

Cells were grown in DMEM media (11995, Thermo Fisher) supplemented with 2 mM GlutaMAX (35050, Thermo Fisher), 10% FBS (VWR), Non-Essential Amino Acids (100 X, 11140, Thermo Fisher), and Penicillin-Streptomycin (100 X, 15070, Thermo Fisher) in 5% $CO_2$ at 37 °C. Huh7 were obtained from UC Berkeley Cell Culture Facility (https://bds.berkeley.edu/facilities/cell-culture#cells) and their identity confirmed through their STR profiling (CELL LINE AUTHENTICATION (ucberkeleydnaseq uencing.com)). Mycoplasma negative status confirmed by PCR Detection Kit.

## Generation of Huh7 cell lines

To generate a UPRER transcriptional reporter, we designed a lentiviral vector that encodes sequence for 5 x UPR response element upstream of a minimal cFos promoter, driving sfGFP (*Adamson et al., 2016*). The sfGFP is fused to a PEST sequence for tighter regulation of the reporter (*Corish and Tyler-Smith, 1999*; *Loetscher et al., 1991*). The vector also allows for constitutive mCherry expression for normalization of the sfGFP signal, and neomycin resistant gene for selection. The transcriptional reporter was then transduced through lentivirus into Cas9-expressing Huh7 cells and selected for using G418 at 800 μg/mL.

## Cell culture supplementation experiments

Cells were plated onto 10 cm plates and allowed to grow overnight. Lipids, tunicamycin, and/or vehicle were directly added to media. Cells were harvested for flow cytometry 16–18 hours after addition of supplements.

## Flow Cytometry

Cells were trypsinized on ice and resuspended in cold FACS buffer (PBS with 0.1% BSA and 2 mM EDTA). Samples were filtered through a 50 μm nylon filter mesh to remove clumps prior to analysis with a five-laser LSR Fortessa (BD Bioscience). Acquired data were analyzed using FlowJo 10.7.2.

## Acknowledgements

We are grateful to all the members of the Dillin lab for intellectual and technical support. We are grateful to the Dernburg lab for use of their Marianas spinning-disc confocal platform. We are thankful to Rebecca A Kohnz for her help with analysis of the lipidomics data. This work was supported by the following grants: GG is supported by T32AG052374, HZ is supported by 2020-A-018-FEL through the Larry L Hillblom Foundation, CKT is supported by F32AG069388 from the National Institute on Aging, RHS is supported by R00AG065200 from the National Institute on Aging, and AD is supported by R01AG059566 from the National Institute on Aging and the Howard Hughes Medical Institute.

Some strains were provided by the CGC, which is funded by the NIH Office of Research Infrastructure Programs (P40 OD010440).

## Additional information

### Funding

| Funder | Grant reference number | Author |
|---|---|---|
| National Institute on Aging | T32AG052374 | Gilberto Garcia |
| Larry L. Hillblom Foundation | 2020-A-018-FEL | Hanlin Zhang |
| National Institute on Aging | F32AG069388 | C Kimberly Tsui |
| National Institute on Aging | R00AG065200 | Ryo Higuchi-Sanabria |
| National Institute on Aging | R01AG059566 | Andrew Dillin |

The funders had no role in study design, data collection and interpretation, or the decision to submit the work for publication.

### Author contributions

Gilberto Garcia, Conceptualization, Data curation, Formal analysis, Validation, Investigation, Visualization, Methodology, Writing - original draft, Writing - review and editing; Hanlin Zhang, Formal analysis, Investigation, Visualization, Methodology, Writing - review and editing; Sophia Moreno, Investigation; C Kimberly Tsui, Resources, Writing - review and editing; Brant Michael Webster, Resources; Ryo Higuchi-Sanabria, Supervision, Investigation, Writing - review and editing; Andrew Dillin, Resources, Supervision, Investigation, Project administration, Writing - review and editing

### Author ORCIDs

Gilberto Garcia http://orcid.org/0000-0001-9959-650X
Hanlin Zhang http://orcid.org/0000-0001-9353-6071
C Kimberly Tsui http://orcid.org/0000-0002-3807-5329
Andrew Dillin http://orcid.org/0000-0002-7427-2629

### Decision letter and Author response

Decision letter https://doi.org/10.7554/eLife.83884.sa1
Author response https://doi.org/10.7554/eLife.83884.sa2

## Additional files

### Supplementary files

• Supplementary file 1. Candidate LD proteins identified by proteome meta-analysis and screen score. Proteins identified in meta-analysis of published LD isolation proteomes (*Zhang et al., 2012*; *Na et al., 2015*; *Vrablik et al., 2015*). Gene description, screen score (scored ±3 in 0.5 increments in comparison to Empty Vector/*sec-11* RNAi control), and approximated developmental stage at time of screen noted. N/A corresponds to genes not screened due to RNAi availability.

• Supplementary file 2. Statistical analysis of tunicamycin survival assay data. Median lifespan, death events counted, and statistics for tunicamycin survival assay of worms grown on Empty Vector (EV) or *let-767* RNAi combined with EV or *sbp-1* RNAi.

• Supplementary file 3. Untargeted lipidomic analysis of *let-767* RNAi treated animals. Normalized values and fold change of identified lipids from *let-767* RNAi treated animals compared to EV controls at day 1 of adulthood. Standard deviation and averages calculated from 6 biological replicates.

• Transparent reporting form

• Source data 1. Unprocessed lipidomic source data of day 1 adult animals. Unprocessed lipidomic analysis data from animals grown on Empty Vector control or let-767 RNAi and collected at day 1 of adulthood.

## Data availability

All data generated or analyzed during this study are included in the manuscript and supporting files; Unprocessed data for Supplementary File 3 has been provided in *Source data 1*.

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
