## [Editor Report]

In this work, the often-surmised but still-poorly understood connection between lipid metabolism and ER stress was explored. The work using genetic techniques and a variety of parallel approaches, implicates key lipid synthetic pathways in the rise and strength of an ER stress signal. The studies as submitted were strong, and create a compelling case for these new connections between lipid metabolism and cellular stress response.

---

## [Decision Letter]

**Decision letter after peer review:**

Thank you for submitting your article "Lipid homeostasis is essential for a maximal ER stress response" for consideration by *eLife*. Your article has been reviewed by 3 peer reviewers, and the evaluation has been overseen by a Reviewing Editor and Vivek Malhotra as the Senior Editor. The reviewers have opted to remain anonymous.

We have now compiled full reviews, and conducted consensus building discussion. Generally, all reviewers agree that the work is unique, interesting, and sufficiently broad in scope to warrant publication in *eLife*, with requested changes. The reviewers' specific comments are listed below, grouped by individual reviewer, with suggested modifications or revisions for a resubmitted version of this important study. Nearly all of the suggestions are to promote clarity in presentation, scholarship, or exposition, and do not require further experimental work. The few suggestions for new experimental studies should be viewed as possible opportunities for expansion of the presented data if such studies have already been executed, although you are welcome to launch new studies according to your temporal, fiscal, and psychological capacities for such endeavor. This is an interesting and thought provoking study, and we appreciate the chance to participate in making it part of the peer-reviewed scientific record.

*Reviewer #1 (Recommendations for the authors):*

a. The experimental system described in the manuscript is powerful as it makes it possible to investigate some important mechanistic questions relatively easily. For example, given the identified ER morphological changes, can overexpression or knockout of the reticulon genes lead to the s-11-activated UPRE-GFP reporter phenotype in acs^-1^ RNAi nematodes? Does RNAi (or overexpression) of such genes recapitulate the let-767 RNAi-induced recovery of ER proteotoxic stress-induced UPRE-GFP in s-11i cells?

b. Comparisons of normalized A.F.U. in Figure 1E and 1G do not seem to correlate well with the images shown in Figure 1D and 1F, respectively. What are these values normalized to? Further explanation is needed on how the normalized A.F.U. was determined from the values obtained with the BioSorter.

Related minor comment

The scale of the y-axis of normalized A.F.U. should be consistent throughout the manuscript. For example, in Figure 4B, normalized A.F.U. is on an X10000 scale while in Figure 4D it is X1000. Both graphs show relatively similar values and differences between WT and let-767 nematodes.

c. Do the cellular levels of LCFA or mmBCFA (iso C17 or oleic acid) and/or levels of iso C17-containing lipids present in the lysate supplement used (Figure 2 and S2) explain the results in let-767 or acs^-1^ RNAi nematodes? Does the recovery occur in a concentration-dependent manner?

d. The ER establishes contact sites with mitochondria. Does the long-chain FA oxidation that occurs in mitochondria have any effect on the hsp-4-GFP levels in either let-767 or acs^-1^ RNAi nematodes?

e. The authors should discuss their findings with respect to the findings published in a recent paper by Bin Liang's lab (i.e., JCB 2021, 220, e202102122 "mmBCFA C17iso ensures endoplasmic reticulum integrity for lipid droplet growth"), as it directly concerns Acs1 involvement in lipid droplet growth in *C. elegans*.

f. One potential addition that could help towards publication in *eLife* might include extending the scope of the manuscript, which would differentiate the work from previously published papers. For example, the experimental system the authors have set up may allow for the testing of the contribution of age, by using young vs. old animals or by generating lysates from either young or old animals. This is not a request for a whole set of new experiments, but determining the impact of age on the contribution of lipids to the UPR might provide a unique and important conclusion that would warrant publication in *eLife*.

*Reviewer #2 (Recommendations for the authors):*

• It should be mentioned either in the title or the abstract that this is in *C. elegans*.

• The paper would be more accessible for readers if the following figure panels were modified to have colorblind friendly colors: Figure S1B, 3A, 3D, 3E, S3A, S3C. This is a good resource for color blind friendly data visualization: https://thenode.biologists.com/data-visualization-with-flying-colors/research/

• In the introduction, the first time a specific gene or pathway is mentioned, it would be helpful to have 1-2 sentences introducing it.

• As a general note on statistics for quantification of the UPR reporter, it says in the figure panels that 3 replicates were used even though the N value is several hundred in most panels. Please clarify what is being counted as 1 replicate. Is the replicate all the larvae from one RNAi experiment? This high N is also why even small impacts on UPR activation have such low p-values throughout the paper. It is sometimes difficult to tell which observations are biologically significant versus statistically significant.

• All microscopy images should include a scale bar.

• Please define "percent mobile" from Figure S3 and how this value was calculated, either in figure legend or in methods section.

• The FRAP data is very interesting, I would not expect that the mobility of an ER luminal protein would be impacted at all. Do you have any idea why this effect is seen?

• Figure S3 is important for the model proposed in Figure 6. I think this figure should be a main figure instead a supplemental figure.

• Was the FRAP experiment performed with a fluorescently tagged Ire1, like mRuby::ire-1b? This would help give credence to the final model in Figure 6 that let-767 knockdown has an impact on Ire1 activation.

• If there is data quantifying UPR expression let-767 RNAi with tunicamycin treatment, that should be included.

• It would be helpful to include an ire-1 RNAi positive control in the tunicamycin surivial curves.

• The genetic screen to identify lipid metabolism genes that crosstalk with ER stress is very interesting! It would be interesting to know why let-767 was selected for follow-up versus the other candidates from Figure S1A.

• It is unclear the exact connection between sbp-1 and let-767 and it would be useful to include more explanation of why sbp-1 knockdown was predicted to alleviate the UPR defect of let-767 knockdown.

• Please expand on the connection between asc-1 and let-767 and speculate on why the phenotype for these two is more similar to that of other LCFA and mmBCFA metabolism genes.

• Images of lipid droplets and ER are difficult to interpret. It would benefit the paper if the paper had quantification of lipid droplet number and size and ratio of ER tubules versus sheets. If this is too time consuming to quantify, it would still be useful to have arrows in the ER images of what is a tube versus a sheet, as this is difficult for readers to determine.

• It would help to understand better the link between HSD17B12 and let-767. Are these homologs? Do they have any sequence similarity?

• Were any experiments performed giving 3-oxoacyl-CoA to *C. elegans* and measuring UPR activation? I understand it could be potentially be converted into a different metabolite, but even so, it would be helpful to include this data if it exists.

• While outside the scope of the paper, it would greatly help in supporting the overall findings if lipidomics were performed in let-767 knockdown worms.

• What are some potential ideas of how let-767 could have impacts on UPR downstream of XBP1 splicing?

*Reviewer #3 (Recommendations for the authors):*

1. Change in lipid droplet (LD) size and number with lysate in let-767 RNAi is minimal. It might be best to report quantification of LD number and size to support their conclusions, such as in Figure 2C-D. It is also difficult to appreciate the difference in ER morphology. I suggest the authors to point out the differences in the panel or to provide images of better quality.

2. There is high background signal for both channels, mCherry and GFP, in Figure 3F. It will be great if the quality of the images can be improved and/or quantification of LD number and size can be added.

3. The authors show that 3- oxoacyl is sufficient to inhibit the UPR activation in huh7 cells. Have the authors similarly tried supplementing *C. elegans* with the fatty acid intermediate 3-oxoacyl?

4. Many of the reported data are from biological duplicates. At least biological triplicates should be performed to ensure reproducibility, unless the practice of biological duplicate is deem acceptable in the field for specific experiments (lifespan).

5. Labelling Figure S3A-B panels with the respective protein tested for mobility will make the figure clearer.

6.P values should be added to Figure 4E-F. It will be good to include the replicates as well with single data point (dot plot superimposed to the bar chart).

7. In the legend of Figure 5B, it should be μM and not uM.

8. In the legend of Figure 5, should it refer to "bar graph of (B)" instead of "H"?

9. In Figure 5B, where is the data for 50 μM Cerulenin?

---

## [Author Response]

Reviewer #1 (Recommendations for the authors):a. The experimental system described in the manuscript is powerful as it makes it possible to investigate some important mechanistic questions relatively easily. For example, given the identified ER morphological changes, can overexpression or knockout of the reticulon genes lead to the s-11-activated UPRE-GFP reporter phenotype in acs^-1^ RNAi nematodes? Does RNAi (or overexpression) of such genes recapitulate the let-767 RNAi-induced recovery of ER proteotoxic stress-induced UPRE-GFP in s-11i cells?

We thank the reviewer for this great suggestion. We have found that treatment of our UPR^ER^ reporter worms with *ret-1* (reticulon protein) RNAi does not induce the UPR^ER^ and does not rescue the UPR^ER^ suppression caused by *let-767* RNAi, as shown in Author response 1. These results are in line with hypothesis that *let-767* RNAi disrupts membrane lipids and not just the morphology of the ER.

**Author response image 1. sa2fig1:** 

b. Comparisons of normalized A.F.U. in Figure 1E and 1G do not seem to correlate well with the images shown in Figure 1D and 1F, respectively. What are these values normalized to? Further explanation is needed on how the normalized A.F.U. was determined from the values obtained with the BioSorter.

The COPAS biosorter allows for quantification of fluorescence normalized to size of the worm using time of flight (a proxy for length) and extinction (a proxy for width). Therefore, sometimes if the worms are different sizes, it may be hard to visualize a significant change in integrated fluorescence intensity by eye, even if a difference exists, whereas the quantification which includes normalization allows more fair assessments across animals of different sizes. This information was in the Materials and methods, but we have also added a sentence to explain the normalization in the figure legends for ease of understanding.

Related minor commentThe scale of the y-axis of normalized A.F.U. should be consistent throughout the manuscript. For example, in Figure 4B, normalized A.F.U. is on an X10000 scale while in Figure 4D it is X1000. Both graphs show relatively similar values and differences between WT and let-767 nematodes.

All sorter images have been made consistent for clarification.

c. Do the cellular levels of LCFA or mmBCFA (iso C17 or oleic acid) and/or levels of iso C17-containing lipids present in the lysate supplement used (Figure 2 and S2) explain the results in let-767 or acs^-1^ RNAi nematodes? Does the recovery occur in a concentration-dependent manner?

This is a very good question. We cannot exclude the possibility that the levels of specific lipids in the lysate may contribute to the differences in rescue results between *let-767* and *acs^-1^* RNAi; however, we believe the levels of oleic acid and iso C17 supplemented in Figure 2—figure supplement 1 are in excess and suggest that the differences in rescue are not due to the levels of these lipids found in the lysate. The text has been modified to clarify these points.

d. The ER establishes contact sites with mitochondria. Does the long-chain FA oxidation that occurs in mitochondria have any effect on the hsp-4-GFP levels in either let-767 or acs^-1^ RNAi nematodes?

Treatment of our UPR^ER^ reporter worms with RNAi of the available Carnatine Palmitoyl Transferase genes (*cpt-1/2/4/5/6*), which are required for the initial step of long-chain FA β-oxidation within the mitochondria, does not have an obvious effect on the GFP levels of the *hsp-4* reporter worms when also treated with *s-11* and/or *let-767* RNAi. While simultaneous knockdown of multiple *cpt* genes may have an increased effect on FA oxidation and increase the amount of long-chain FAs available, we do not expect an increase in available long-chain FAs to impact our phenotype as supplementation of excess oleic acid (a long-chain FA) did not rescue the UPR^ER^ suppression caused by *let-767* RNAi. See Author response image 2.

e. The authors should discuss their findings with respect to the findings published in a recent paper by Bin Liang's lab (i.e., JCB 2021, 220, e202102122 "mmBCFA C17iso ensures endoplasmic reticulum integrity for lipid droplet growth"), as it directly concerns Acs1 involvement in lipid droplet growth in *C. elegans*.

We thank the reviewer for bringing up this important citation and the revised manuscript references this paper throughout the manuscript and within the discussion.

f. One potential addition that could help towards publication in eLife might include extending the scope of the manuscript, which would differentiate the work from previously published papers. For example, the experimental system the authors have set up may allow for the testing of the contribution of age, by using young vs. old animals or by generating lysates from either young or old animals. This is not a request for a whole set of new experiments, but determining the impact of age on the contribution of lipids to the UPR might provide a unique and important conclusion that would warrant publication in eLife.

We have significantly expanded the discussion to highlight the importance of lipid pathways in aging studies. While we believe that the experiments proposed by the reviewer are excellent, we believe these are out of scope of the current manuscript and better suited for a follow-up study but have written a comprehensive section on this topic in the discussion.

Reviewer #2 (Recommendations for the authors):• It should be mentioned either in the title or the abstract that this is in *C. elegans*.

This is a great point and the abstract now highlights that this work is done in *C. elegans*.

• The paper would be more accessible for readers if the following figure panels were modified to have colorblind friendly colors: Figure S1B, 3A, 3D, 3E, S3A, S3C. This is a good resource for color blind friendly data visualization: https://thenode.biologists.com/data-visualization-with-flying-colors/research/

We thank the reviewer for this excellent suggestion and all figures have now been changed to colorblind friendly colors using Okabe_Ito.

• In the introduction, the first time a specific gene or pathway is mentioned, it would be helpful to have 1-2 sentences introducing it.

We have now added a sentence introducing every gene upon its first mention.

• As a general note on statistics for quantification of the UPR reporter, it says in the figure panels that 3 replicates were used even though the N value is several hundred in most panels. Please clarify what is being counted as 1 replicate. Is the replicate all the larvae from one RNAi experiment? This high N is also why even small impacts on UPR activation have such low p-values throughout the paper. It is sometimes difficult to tell which observations are biologically significant versus statistically significant.

This is a good point considering the sample size are quite large in our biosorter assays. We have now added clarifying text to every figure legend stating that representative data one of three biological replicates are shown. Generally, our data show very significant changes in values and we believe that there are both biologically and statistically significant observations.

• All microscopy images should include a scale bar.

We apologize for this oversight and scale bars have been included to all microscopy figures.

• Please define "percent mobile" from Figure S3 and how this value was calculated, either in figure legend or in methods section.

Percent mobile has been defined in the text and its calculation summarized in the FRAP analysis methods section.

• The FRAP data is very interesting, I would not expect that the mobility of an ER luminal protein would be impacted at all. Do you have any idea why this effect is seen?

This is a great question. We hypothesize that this effect is likely a result of the overall altered structure of the ER and its membrane leading to novel interaction or crowding within the lumen. Our hypothesis has been clarified within the text.

• Figure S3 is important for the model proposed in Figure 6. I think this figure should be a main figure instead a supplemental figure.

Thank you for this great suggestion. Figure 5—figure supplement 1 has been relabeled as a main figure.

• Was the FRAP experiment performed with a fluorescently tagged Ire1, like mRuby::ire-1b? This would help give credence to the final model in Figure 6 that let-767 knockdown has an impact on Ire1 activation.

This is a great suggestion. We have performed a FRAP experiment on an IRE-1B::mRuby and indeed, *let-767* RNAi influenced its dynamics; however, we saw an increase in IRE-1B mobility as shown in Author response image 3. However, due to IRE-1B is missing its luminal domain, so we are uncertain of what conclusions can be made of this since it is entirely possible that dynamics of a truncated IRE-1B would not mimic a full-length protein. Therefore, we opted to exclude this data from the manuscript.

**Author response image 3. sa2fig3:** 

• If there is data quantifying UPR expression let-767 RNAi with tunicamycin treatment, that should be included.

This is a great suggestion. Unfortunately, we currently no longer have access to a biosorter, but were able to do an imaging experiment as shown in Author response image 4. In agreement with our RNAi induced ER stress data, we observed a reduction in intestinal *hsp-4* induction when animals grown on *let-767* RNAi were treated with tunicamycin. Additionally, *tag-335* RNAi targets the n-glycosylation pathway and would be functionally similar to tunicamycin treatment, which also blocks N-linked glycosylation. Quantification of *tag-335/let-767* RNAi has also been provided in Figure 1B-C.

**Author response image 4. sa2fig4:** 

• It would be helpful to include an ire-1 RNAi positive control in the tunicamycin surivial curves.

In our tunicamycin assays, we provide DMSO negative controls (Figure 3F), which are done in parallel with the tunicamycin survival assays in Figure 4E. Since we see that tunicamycin treatment severely impacts lifespan compared to a DMSO control, we are confident that our tunicamycin treatment is working and thus do not believe an additional positive control is necessary in this assay, especially since this is a commonly used protocol published.

• The genetic screen to identify lipid metabolism genes that crosstalk with ER stress is very interesting! It would be interesting to know why let-767 was selected for follow-up versus the other candidates from Figure S1A.

This is a great point and we have expanded our rationale for the focus on *let-767*.

• It is unclear the exact connection between sbp-1 and let-767 and it would be useful to include more explanation of why sbp-1 knockdown was predicted to alleviate the UPR defect of let-767 knockdown.

Another great point and we have expanded the link between *let-767* and *sbp-1*.

• Please expand on the connection between asc-1 and let-767 and speculate on why the phenotype for these two is more similar to that of other LCFA and mmBCFA metabolism genes.

We have also expanded on the similarities between *acs^-1^* and *let-767* in the discussion.

• Images of lipid droplets and ER are difficult to interpret. It would benefit the paper if the paper had quantification of lipid droplet number and size and ratio of ER tubules versus sheets. If this is too time consuming to quantify, it would still be useful to have arrows in the ER images of what is a tube versus a sheet, as this is difficult for readers to determine.

We understand that the previous images were challenging and have now added higher quality images to see differences more easily and have also added arrows to highlight differences in lipid droplets and ER features that we focus on.

• It would help to understand better the link between HSD17B12 and let-767. Are these homologs? Do they have any sequence similarity?

The link between HSD17B12 and let-767 have been more clearly explained within the text.

• Were any experiments performed giving 3-oxoacyl-CoA to *C. elegans* and measuring UPR activation? I understand it could be potentially be converted into a different metabolite, but even so, it would be helpful to include this data if it exists.

We have tried supplementing 3-oxoacyl and found that it had a statistically significant, but only had a mild effect on our reporter, as is shown in Author response image 5. This is potentially due to the intermediate being metabolized by bacteria or insufficient availability due to insolubility of the intermediate. Since it is hard to make conclusions of this data, we did not include this in the manuscript.

**Author response image 5. sa2fig5:** 

• While outside the scope of the paper, it would greatly help in supporting the overall findings if lipidomics were performed in let-767 knockdown worms.

This is a great suggestion and we have put enormous effort into performing lipidomics in the worm. Unfortunately, we did not see specific causal lipids We have included lipidomics data for worms treated with let-767 RNAi (TableS3) and discussed a possible reason for a specific causal lipid not being identified and only saw a general reduction in most lipids. This information has been added and descriptions were added both to the results and discussions.

• What are some potential ideas of how let-767 could have impacts on UPR downstream of XBP1 splicing?

This is a great question and we have expanded on/clarified our ideas of how let-767 RNAi might impact xbp-1 downstream of splicing.

Reviewer #3 (Recommendations for the authors):1. Change in lipid droplet (LD) size and number with lysate in let-767 RNAi is minimal. It might be best to report quantification of LD number and size to support their conclusions, such as in Figure 2C-D. It is also difficult to appreciate the difference in ER morphology. I suggest the authors to point out the differences in the panel or to provide images of better quality.

Higher quality images have been added and arrows added to highlight the differences in LDs and ER.

2. There is high background signal for both channels, mCherry and GFP, in Figure 3F. It will be great if the quality of the images can be improved and/or quantification of LD number and size can be added.

Similar to point 1, higher quality images have been added to Figure 3F and arrows added to highlight LDs and ER features.

3. The authors show that 3- oxoacyl is sufficient to inhibit the UPR activation in huh7 cells. Have the authors similarly tried supplementing *C. elegans* with the fatty acid intermediate 3-oxoacyl?

This is a great question and we have tried supplementing 3-oxoacyl and found that it had a statistically significant, but only had a mild effect on potentially due to the intermediate being metabolized by bacteria or insufficient availability due to insolubility of the intermediate. Due to these caveats, we believe the data are inconclusive and opted to not include them in the manuscript, although it is provided in Author response image 6.

**Author response image 6. sa2fig6:** 

4. Many of the reported data are from biological duplicates. At least biological triplicates should be performed to ensure reproducibility, unless the practice of biological duplicate is deem acceptable in the field for specific experiments (lifespan).

Almost all experiments have 3 or more biological replicates except the following:

Figure 1—figure supplement 1F: A third replicate was performed during the revision of the manuscript.FRAP Figure 4: Data are pooled from two repeats; unfortunately, we no longer have access to the microscope where we performed this assay and it is highly labor intensive. However, our two replicates show virtually identical data and we are confident in the reproducibility.Figure 5—figure supplement 1: Although we no longer have access to a biosorter, we have performed three additional replicates via imaging during the revision of the manuscript.Figure 6B: a third replicate has been performed during the revision of the manuscript.

5. Labelling Figure S3A-B panels with the respective protein tested for mobility will make the figure clearer.

FRAP curve panels have been re-labeled for clarity.

6.P values should be added to Figure 4E-F. It will be good to include the replicates as well with single data point (dot plot superimposed to the bar chart).

P values and replicate data points have been added to Figure 4E-F.

7. In the legend of Figure 5B, it should be μM and not uM.

We apologize for the oversight and the legend text has been corrected.

8. In the legend of Figure 5, should it refer to "bar graph of (B)" instead of "H"?

The legend text has been corrected.

9. In Figure 5B, where is the data for 50 μM Cerulenin?

The data for Cerulenin has been added.